# Human Attribute Recognition—
# A Comprehensive Survey

**Ehsan Yaghoubi** [1,*,†] ⓘ, **Farhad Khezeli** [2], **Diana Borza** [3], **SV Aruna Kumar** [4] **and João Neves** [5]
**and Hugo Proença** [1,*]

1    IT: Instituto de Telecomunicações, University of Beira Interior, 6201-001 Covilhã, Portugal
2    Science and Research Branch, Islamic Azad University, Tehran 1477893855, Iran; farhad.khezeli@srbiau.ac.ir
3    Faculty of Computer Science, Technical University of Cluj-Napoca, 400114 Cluj-Napoca, Romania ;
     diana.borza@cs.utcluj.ro
4    Faculty of Computer Science, University of Beira Interior, 6201-001 Covilhã, Portugal;
     aruna.venkateshappa@ubi.pt
5    TomiWorld, 3500-106 Viseu, Portugal; JoaoNeves@tomiworld.com
*    Correspondence: Ehsan.yaghoubi@ubi.pt (E.Y.); hugomcp@di.ubi.pt (H.P.); Tel.: +351-915840758 (E.Y.)
†    Current address: SOCIA Lab., Faculty of Computer Science, University of Beira Interior, Portugal.

**Abstract:**    Human Attribute Recognition (HAR) is a highly active research field in computer vision and pattern recognition domains with various applications such as surveillance or fashion. Several approaches have been proposed to tackle the particular challenges in HAR. However, these approaches have dramatically changed over the last decade, mainly due to the improvements brought by deep learning solutions. To provide insights for future algorithm design and dataset collections, in this survey, (1) we provide an in-depth analysis of existing HAR techniques, concerning the advances proposed to address the HAR's main challenges; (2) we provide a comprehensive discussion over the publicly available datasets for the development and evaluation of novel HAR approaches; (3) we outline the applications and typical evaluation metrics used in the HAR context.

**Keywords:**    human attribute recognition; imbalanced learning; pedestrian recognition; privacy concerns; clothing attributes; soft biometrics; appearance-based learning; deep learning

---

## 1. Introduction

Over recent years, the increasing amount of multimedia data available in the Internet or supplied by CCTV devices deployed in public/private environments has been raising the requirements for solutions able to automatically analyse human appearance, features and behavior. Hence, Human Attribute Recognition (HAR) has been attracting increasing attentions in the computer vision/pattern recognition domains, mainly due to its potential usability for a wide range of applications (e.g., crowd analysis [1], person search [2,3], detection [4], tracking [5], and re-identification [6]). HAR aims at describing and understanding the subjects' traits (such as their hair color, clothing style [7], gender [8], etc.) either from full-body or facial data [9]. Generally, there are four main sub-categories in this area of study:

- Facial Attribute Analysis (FAA). Facial attribute analysis aims at estimating the facial attributes or manipulating the desired attributes. The former is usually carried out by extracting a comprehensive feature representation of the face image, followed by a classifier to predict the face attributes. On the other hand, in manipulation works, face images are modified (e.g., glasses are removed or added) using generative models.

- Full-body Attribute Recognition (FAR). Full-body attribute recognition regards the task of inferring the soft-biometric labels of the subject, including clothing style, head-region attributes, recurring actions (talking to the phone) and role (cleaning lady, policeman), regardless of the location or body position (eating in a restaurant).
- Pedestrian Attribute Recognition (PAR). As an emerging research sub-field of HAR, it focuses on the full-body human data that have been exclusively collected from video surveillance cameras or panels, where persons are captured while walking, standing, or running.
- Clothing Attribute Analysis (CAA). Another sub-field of human attribute analysis that is exclusively focused on clothing style and type. It comprises several sub-categories such as in-shop retrieval, costumer-to-shop retrieval, fashion landmark detection, fashion analysis, and cloth attribute recognition, each of which requires specific solutions to handle the challenges in the field. Among these sub-categories, cloth attribute recognition is similar to pedestrian and full-body attribute recognition and studies the clothing types (e.g., texture, category, shape, style).

The typical pipeline of the HAR systems is given in Figure 1, which indicates the requirement of a dataset preparation prior to designing a model. As shown in Figure 1, preparing a dataset for this problem typically comprises four steps:

1. Capturing raw data, which can be accomplished using mobile cameras (e.g., drone) or stationary cameras (e.g., CCTV). Also, the raw data might even be collected from images/videos publicly available (e.g., *Youtube*, or similar sources).
2. In most supervised training approaches, HAR models consider one person at a time (instead of analyzing a full-frame with multiple persons). Therefore, detecting the bounding boxes of each subject is essential and can be done by state-of-the-art object detection solutions (i.e., Mask R-CNN [10], You Only Look Once (YOLO) [11], Single Shot Detection (SSD) [12], etc.)
3. If the raw data is in video format, spatio-temporal information should be kept. in such cases, the accurate tracking of each object (subject) in the scene can significantly ease the annotation process.
4. Finally, in order to label the data with semantic attributes, all the bounding boxes of each individual are displaced to human annotators. based on human perception, the desired labels (e.g., 'gender' or 'age') are then associated to each instance of the dataset.

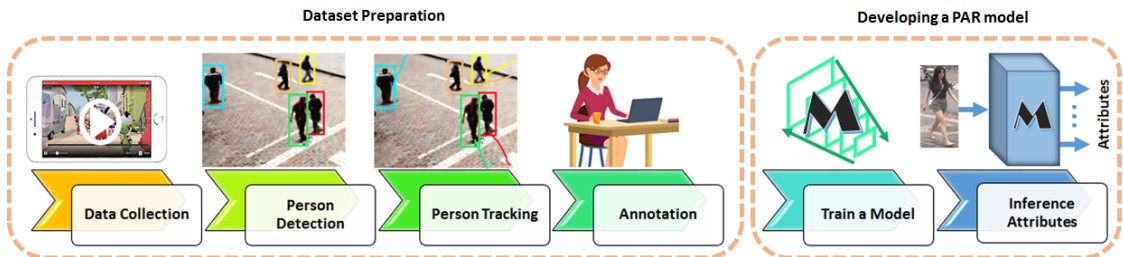

**Figure 1.** Typical pipeline to develop a HAR model.

Regarding the data-type and available annotations, there are many possibilities for designing HAR models. Early researches were based on crafted feature extractors. Typically, the linear Support Vector Machine (SVM) was used with different descriptors (such as ensemble of localized features, local binary patterns, color histograms, histogram of oriented gradients) to estimate the human attributes. However, as the correlation between human attributes were ignored in traditional methods, one single model was not suitable for estimating several attributes. For instance, descriptors suitable for gender recognition could not be effective enough to recognize the hairstyle. Therefore, conventional methods mostly focused on obtaining independent feature extractors for each attribute. After the advent of Convolutional Neural Networks (CNNs) and using it as a holistic feature extractor, a growing number of methods focused on models that can estimate multiple attributes at once. Earlier deep-based

methods used shallow networks (e.g., 8-layer AlexNet [13]), while later models moved towards deeper architectures (e.g., residual network (ResNet)) [14].

The difficulties in HAR originates mainly due to the high-variability in human appearance particularly in intra-class samples. Nevertheless, the following factors have been identified as the basis for the development of robust HAR systems:

- learn in an end-to-end manner and yield multiple attributes at once;
- extract a discriminative and comprehensive feature representation from the input data;
- leverage the intrinsic correlations between attributes;
- consider the location of each attribute in a weakly supervised manner;
- are robust to primary challenges such as low-resolution data, pose variation, occlusion, illumination variation, and cluttered background;
- handle the classes imbalance;
- manage the limited-data problem effectively.

Despite the relevant advances and many research articles published, HAR can be considered still in its early stages. For the community to come up with original solutions, it is necessary to be aware of the history of advancements, state-of-the-art performance, and the existing datasets related to this field. Therefore, in this study, we discuss a collection of HAR related works, starting from the traditional one to the most recent proposals, and explain their possible advantages/drawbacks. We further analyze the performance of recent studies. Moreover, although we identified more than 15 publicly available HAR datasets, to the best of our knowledge, we do not have a clear discussion on the aspects that one should observe while collecting a HAR dataset. Thus, after taxonomizing the datasets and describing their main features and data collection setups, we discuss the critical issues of the data preparation step.

Regarding the previously published surveys that addressed similar topics, we particularly mention Zheng et al. [15], where the facial attribute manipulation and estimation methods have been reviewed. However, to date, there is no solid survey on the recent advances in other sub-categories of human attribute analysis. As the essence of full-body, pedestrian, and cloth attribute recognition methods are similar to each other; in this paper, we cover all of them with a particular focus on the pedestrian attribute recognition methods. Meanwhile, Reference [16] is the only work similar to our survey that is about pedestrian attribute recognition. Several points distinguish our work from Reference [16]:

- The recent literature on HAR has been mostly focused on addressing some particular challenges of this problem (such as class imbalance, attribute localization, etc.) rather devising a general HAR system. Therefore, instead of providing a methodological categorization of the literature as in Reference [16], our survey proposes a challenge-based taxonomy, discussing the state-of-the-art solutions and the rationale behind them;
- Contrary to Reference [16], we analyze the motivation of each work and the intuitive reason for its superior performance;
- The datasets main features, statistics and types of annotation are compared and discussed in detail;
- Beside the motivations, we discuss HAR applications, divided into three main categories: security, commercial, and related research directions.

*Motivation And Applications*

Human attribute recognition methods extract semantic features that describe human-understandable characteristics of the individuals in a scene, either from images or video sequences, ranging from demographic information (gender, age, race/ethnicity), appearance attributes (body weight, face shape, hairstyle and color etc.), emotional state, to the motivation and attention of people (head pose, gaze direction). As they provide vital information about humans,

such systems have already been integrated into numerous real-world applications, and are entwined with many technologies across the globe.

Indisputably, HAR is one of the most important steps in any visual surveillance system. Biometric identifiers are extracted to identify and distinguish between the individuals. Based on the biometric traits, humans are uniquely identified, either based on their facial appearance [17–19], iris patterns [20] or on behavioral traits (gait) [21,22]. With the increase of surveillance cameras worldwide, the research focus has shifted from (hard-)biometric (iris recognition, palm-print) to soft biometric identifiers. The latter describe human characteristics, taxonomized into a humanly understandable manner, but are not sufficient to uniquely differentiate between individuals. Instead, they are descriptors used by humans to categorize their peers into several classes.

On a top level, HAR applications can be divided into three main categories: *security and safety*, *research* directions, and *commercial applications*.

Yielding high-level semantic information, HAR could provide auxiliary information for different computer vision tasks, such as person re-identification ([23,24]), human action recognition [25], scene understanding, advanced driving assistance systems, and event detection ([26]).

Another fertile field where HAR could be applied is in human drone surveillance. Drones or Unmanned Aerial Vehicles (UAV), although initially designed for military applications, are rapidly extending to various other application domains, due to their reduced size, swiftness, and ability to navigate through remote and dangerous environments. Researchers in multiple fields have started to use UAVs drones in their research work, and, as a result, the Scopus database has shown an increase in the papers related to UAVs, from 11 ($4.7 \times 10^6$ of total papers) papers published in 2009 to 851 ($270.0 \times 10^6$ of total articles) published in 2018 [27]. In terms of human surveillance, drones have been successfully used in various scenarios, ranging from rescue operations and victim identification, people counting and crowd detection, to police activities. All these applications require information about human attributes.

Nowadays, researchers in universities and major car industries work together to design and build the self-driving cars of the future. HAR methods have important implications in such systems as well. Although numerous papers addressed the problem of pedestrian detection, pedestrian attribute recognition is one of the keys to future improvements. Cues about the pedestrians' body and head orientation provide insights about their intent, and thus avoiding collisions. The pedestrians' age is another aspect that should be analyzed by advanced driving assistance systems to decrease vehicle speed when children are on the sidewalk. Finally, other works suggest that even pedestrians' accessories could be used to avoid collisions: starting from the statistical evidence that collisions between pedestrians and vehicles are more frequent on rainy days, in Reference [28] authors suggest that detecting whether a pedestrian has on open umbrella could reduce traffic incidences.

As mentioned above, the applications of biometric cues are not limited to surveillance systems. Such traits have necessary implications also in commercial applications (logins, medical records management) and government applications (ID cards, border, and passport control) [29]. Also, a recent trend is to have advertisement displays in malls and stores equipped with cameras and HAR systems to extract socio-demographic attributes of the audience and present appropriate and targeted ads based on the audience's gender, generation or age.

Of course, this application list is not exhaustive, and numerous other practical uses of HAR can be envisioned, as this task has implications in all fields interested in and requiring (detailed) human description.

In the remainder of this paper, we first describe the HAR preliminaries—dataset preparation, and the general difference between the earliest and most recent model approaches. In Section 3, we survey the HAR techniques from their main challenge point-of-view, in order to increase the reader's creativity in introducing novel ideas for solving the task of HAR. Further, in Sections 4 and 5, we detail the existing PAR, FAR, and CAA datasets and commonly used evaluation metrics for HAR

models. In Section 6, we discuss the advantages and disadvantages of the above-presented methods and compare their performance over the well-known HAR datasets.

## 2. Human Attribute Recognition Preliminaries

To recognize the human full-body attributes, it is necessary to follow a two-step pipeline, as depicted in Figure 1. In the remainder of this section, each of these steps is described in detail.

### 2.1. Data Preparation

Developing a HAR model requires relevant annotated data, such that each person is manually labeled based on its semantic attributes. As discussed in Section 4, there are different types of data sources such as fashion, aerial, and synthetic datasets, which could be collected from the Internet resources (e.g., Flickr) or through static or mobile cameras in indoor/outdoor locations. HAR models are often developed to recognize human attributes from person bounding boxes (instead of analyzing an entire frame comprising multiple persons). That is why, after the data collection step, it is required to pre-process the data and extract the bounding box of each person. Earlier methods use human annotators to specify the person locations in each image, and then assign soft biometric labels to each of person bounding boxed, while recent approaches take advantage of the CNN-based person detectors (e.g., Reference [10])—or trackers [30], if the data is collected as videos—to provide the human annotators with person bounding boxes for more labeling processes. We refer the interested reader to Reference [31] for more information on person detection and tracking methods.

### 2.2. HAR Model Development

In this part, we discuss the main problem in HAR and highlight the differences between the earlier methods and the most recent deep learning-based approaches.

In machine learning, classification is most often seen as a supervised learning task, in which a model learns from the labeled input data to predict the appeared classes in the unseen data. For example, given many person images with gender labels ('male' or 'female'), we develop an algorithm to find the relationship between images and labels, based on which we predict the labels of the new images. Fisher's linear discriminant [32], support vector machine [33], decision trees [34,35], and neural networks [36,37] are examples of classification algorithms. As the input data is large or suspected to have redundant measures, before analyzing it for classification, the image is transformed into a reduced set of features. This transformation can be performed using neural networks [38] or different feature descriptors [39]—such as Major Colour Spectrum Histogram (MCSH) [40], Color Structure Descriptor (CSD) [41,42], Scale Invariant Feature Transform (SIFT) [43,44], Maximally Stable Colour Regions (MSCR) [45,46], Recurrent Highly-Structured Patches (RHSP), and Histogram of Oriented Gradients (HOG) [47–49]. Image descriptors are not generalized to all the computer vision problems and may be suitable only for specific data type—for example, color descriptors are only suitable for color images. Therefore, models based on feature descriptors are often called hand-crafted methods, in which we should define and apply proper feature descriptors to extract a comprehensive and distinct set of features from each input image. This process may require more feature engineering, such as dimensionality reduction, feature selection, and fusion. Later, based on the extracted features, multiple classifiers are learned, such that each one is specialized in predicting specific attributes of the given input image. As the reader may have noticed, these steps are offline (the result of each step should be saved on the disk as the input of the next step). On the contrary, deep neural networks are capable of modeling the complex non-linear relationships between the input image and labels, such that the feature extraction and classifier learning are performed simultaneously. Deep neural networks are implemented as multi-level (large to small feature-map dimensions) layers, in which different processing filters are convoluted with the output of the previous layer. In the first levels of the model, low-level features (e.g., edges) are extracted, while mid-layers and last-layers extract the mid-level features (e.g., texture) and high-level features (e.g., expressiveness

of the data), respectively. To learn the classification, several fully connected layers are added on top of the convolutional layers (known as a backbone) to map the last feature map to a feature vector with several neurons equal to the number of class labels (attributes).

Several major advantages of deep learning approaches moved the main research trend towards the deep neural network methods. First, CNNs are end-to-end (i.e., both the feature extraction and classification layers are trained simultaneously). Second, the deep neural networks' high generalization ability has provided the possibility of transferring the knowledge of other similar fields to scenarios with limited data. As an example, applying the weights of a model that has been trained on a large dataset (e.g., ImageNet [50]) not only has shown positive effects on the accuracy of the model but also has decreased the convergence time and over-fitting problem [51–53]. Thirdly, CNNs could be designed to handle multiple tasks and labels in a unified model [54,55].

To fully understand the discussion on the state of the arts in HAR, we encourage the newcomer readers to read about different architectures of deep neural networks and their components in References [56,57]. Meanwhile, common evaluation metrics are explained in Section 5.

## 3. Discussion of Sources

As depicted in Figure 2, we identified five major challenges frequently addressed by the literature on HAR—localization, limited learning data, attribute relation, body-part occlusion, and data class imbalance.

HAR datasets only provide the labels for a bounding box of person, but the locations related to each attribute are not annotated. Finding which features are related to which parts of the body is not a trivial task (mainly because body posture is always changing), and not fulfilling it may cause an error in prediction. For example, recognizing the 'wearing sunglasses' attribute in a full-body image of a person without considering the eyeglasses' location may lead to omitting the sunglasses feature information due to extensive pooling layers and a small region of the eyeglasses, compared to the whole image. This challenge is known as localization (Section 3.1), as in which we attempt to extract features of different spatial locations of the image to be certain no information is lost, and we can extract distant features from the input data.

Earlier methods used to work with limited data as the mathematical calculations were computationally expensive, and increasing the amount of data could not justify the exponential computational cost and the amount of improvement in the accuracy. After the deep learning breakthrough, more data proved to be effective in the generalization ability of the models. However, collecting and annotating very large datasets is prohibitively expensive. This issue is known as limited data challenge, which has been the subject of many studies in the deep neural network fields of study, including deep-based HAR, addressed in Section 3.2.

In the context of HAR, dozens of attributes are often analyzed together. As humans, we know that some of these attributes are highly correlated, and knowing one can improve the recognition probability of the other attributes. For example, for a person wearing a 'tie,' it is less likely to wear a 'Pyjama' and more likely to wear a 'shirt' and 'suit'. Studies that address the relationship between attributes as their main contribution are categorized in the 'attribute relation' taxonomy and discussed in Section 3.3.

Body parts occlusion is another challenge when dealing with HAR data that has not yet been addressed by many studies. The challenge in occluded body parts is not only about the missing information of the body parts, but also the presence of some misleading features of other persons or objects. Further, because in HAR, some attributes are related to specific regions, considering the occluded parts before the prediction is important. For example, for a person with an occluded lower body, yielding predictions about the attributes located in the lower body region is questionable. In Section 3.4, we discuss the methods and ideas that have particularly addressed the occlusion in HAR data.

Another critical challenge in HAR is the imbalanced number of samples in each class of data. Naturally, an observer sees fewer persons wearing long coats, while there are many persons in the community that appear with a pair of jeans. That is why the HAR datasets are intrinsically imbalanced and cause the model to be biased/over-fitted on some classes of data. Many studies address this challenge in HAR data, which have been discussed in Section 3.5.

Among the major challenges in HAR, considering attribute correlation and extracting fine-grained features from local regions of the given data have attracted the most attention, such that recent works [58,59] attempt to develop some models that could address both challenges at the same time. Data class imbalance is another contribution of many HAR methods which is often handled by applying weighted loss functions to increase the importance of the minority samples and decrease the effect of the samples from classes with many samples. To deal with limited data challenges, scholars frequently apply the existing holistic transfer learning and augmentation techniques in computer vision and pattern recognition. In this section, we discuss the significant contributions of the literature works in alleviating the main challenges in HAR.

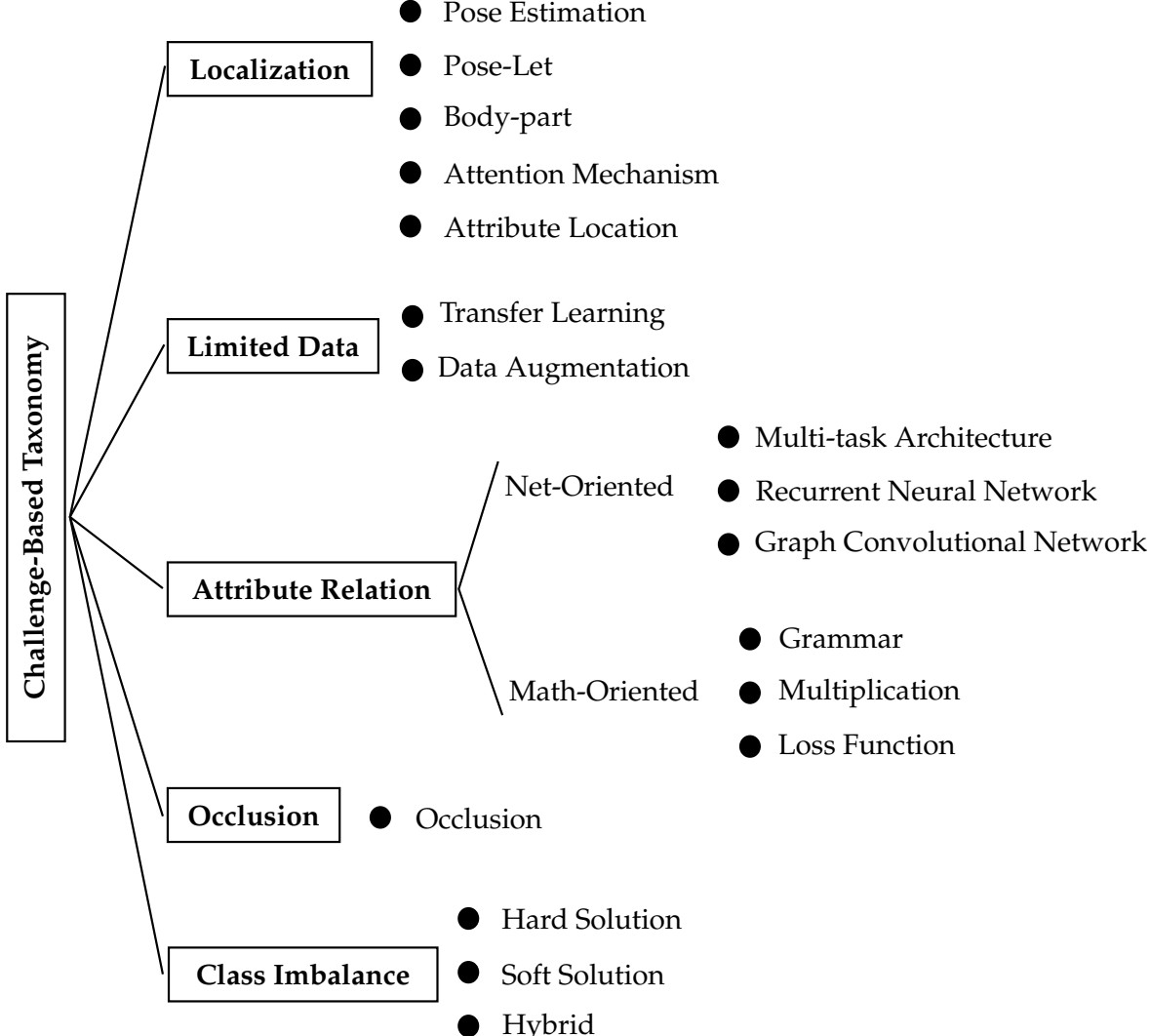

**Figure 2.** The proposed taxonomy for main challenges in HAR.

*3.1. Localization Methods*

Analyzing human full-body images only yields the global features; therefore, to extract distinct features from each identity, analyzing local regions of the image becomes important [60]. To capture the human fine-grained features, typical methods divide the person's image into several strides or patches

and aggregate all the decisions on parts to yield the final decision. The intuition behind these method is that, decomposition of human-body and comparing it with others is intuitively similar to localizing the semantic body-parts and then describing them. In the following, we survey 5 types of localization approaches—(1) attribute location-based methods that consider the spatial location of each attribute in the image (e.g., glasses features are located in the head area, while shoes features are in the lower part of the image); (2) attention mechanism-based techniques that attempts to automatically find on the most important locations of the image based on the ground truth labels; (3) body part-based models, in which the model first locates the body parts (i.e., head, torso, hands, and legs) and then extract the related features from each body parts and aggregate them; (4) pose-let-based techniques that extracts the features from many random locations of the image and aggregate them; (5) pose estimation-based methods that use the coordination of the body skeleton/joints to extract the local features.

### 3.1.1. Pose Estimation-Based Methods

Considering the effect of the body-pose variation of the feature representation, [61] proposes to learn multiple attribute classifiers so that each of them is suitable for a specific body-pose. Therefore, authors use the Inception architecture [62] as the backbone feature extractor, followed by three branches to capture the specific features of the front, back, and side views of the individuals. Simultaneously, a view-sensitive module analyzes the extracted features from the backbone to refine each branch's scores. The final results are the concatenation of all the scores. Ablation studies on the PETA dataset show that a plain Inception model achieves an 84.4 *F*1-score, while for the model with a pose-sensitive module, this metric increases to 85.5.

Reference [63] is another research that takes advantage of pose estimation for improving the performance of pedestrian attribute recognition. In this work, Li et al. suggested a two-stream model whose results are fused, allowing the model to benefit from both regular global and pose-sensitive features. Given an input image, the first stream extracts the regular global features. The pose-sensitive branch comprises three steps—(1) coarse pose estimator (body-joint coordinates predictor) by applying the approach proposed in Reference[64], (2) region localization that uses the body-pose information to spatially transform the desired region, originally proposed in References [65], (3) fusion layer that concatenates the features of each region. In the first step, pose coordinates are extracted to be shared with the second module, in which body parts are localized by using spatial transformer networks [65]. A specific classifier is then trained for each region. Finally, the extracted features from both streams are concatenated to return a comprehensive feature representation of the given input data.

### 3.1.2. Pose-Let-Based Methods

The main idea of pose-let based methods is to provide a bag-of-features from the input data using different patching technique. As earlier methods lacked accurate body part detectors, overlapping patches of the input images were used to extract local features.

Reference [66] is one of the first techniques in this group that uses Spatial Pyramid Representation (SPR) [67] to divide the images into grids. Unlike a standard bag-of-features method that extracts the features from a uniform patching distribution, they suggest a recursive splitting technique, in which each grid has a parameter that is jointly learned with the weight vector. Intuitively, the spatial grids are varying for each class, which leads to better feature extraction.

In Reference [68], hundreds of pose-lets are detected from the input data; a classifier is trained for each pose-let and semantic attribute. Then, another classifier aggregates the body-part information, with emphasis on the pose-lets taken from usual viewpoints that have discriminative features. A third classifier is then used to consider the relationship between the attributes. This way, by using the obtained feature representation, the body pose and viewpoint are implicitly decomposed.

Noticing the importance of accurate body-part detection when dealing with clothing appearance variations, Reference [69] proposes to learn a comprehensive dictionary that considers various appearance part types (e.g., representing the lower-body in different appearances from bare legs

to long skirts). To this end, all the input images are divided into static overlapping cells, each of which is represented by a feature descriptor. Then, as a result of feature clustering into *K* clusters, they represent *k* types of appearance parts.

In Reference [70], the authors targeted the human attributes and action recognition from still images. To this end, supposing that the available human bounding boxes are located in the center of the image, the model learns the scale and positions of a series of image partitions. Later, the model predicts the labels based on the reconstructed image from the learned partitions.

To address the large variation in articulation, angle, and body-pose [71] proposes a CNN-based features extractor, in which each pose-let is fed to an independent CNN. Then, a linear SVM classifier learns to distinguish the human attributes based on the aggregation between the full-body and pose-let features.

References [72,73] showed that not only CNNs can yield a high-quality feature representation from the input, but also they are better at classification than SVM classifiers. In this context, Zhu et al. propose to predict multiple attributes at-once, by implicit regard to the attribute dependencies. Therefore, the authors divide the image into 15 static patches and analyze each one with a separate CNN. To consider the relationship between attributes and patches, they connect the output of some specific CNNs to the relevant static patches. For example, the upper splits of the images are connected to the head and shoulder's attributes.

Reference [74] claims that in previous pose-let works, the location information of the attributes is ignored. For example, to recognize whether a person wears a hat or not, knowing that this feature is related to the upper regions of the image can guide the model to extract more relevant features. To implement this idea, the authors used an Inception [62] structure, in which the features of three different levels (low, middle, and high levels) are fed to three identical modules. These modules extract different patches from the whole and part of the input feature maps. The aggregation of the three branches yields the final feature representation. By following this architectural design, the model implicitly learns the regions related to each attribute in a weakly supervised method. Surprisingly, the baseline (the same implementation without the proposed module) achieves better results on the PETA dataset (84.9 vs. 83.4 of $F1$), while on RAP dataset, the results of the model equipped with their proposed module ($F168.6$) is better with a margin of 2.

Reference [75] receives the full frames and uses the scene features (i.e., hierarchical contexts) to help the model learn the attributes of the targeted person. For example, in a sports scene, it is expected that people have sporty style clothing. Using Fast R-CNN [76], the bounding box of each individual is detected, and several pose-let are extracted. After feeding the input frame and its Gaussian pyramids into several convolutional layers, four fully connected branches are added to the top of the network to yield four scores (from human bounding box, pose-lets, nearest neighbors of the selected parts, and full-frame) for a final concatenation.

### 3.1.3. Part-Based Methods

Extracting discriminative fine-grained features often requires first to localize patches of the relevant regions in the input data. Unlike pose-let-based methods that detect the patches from the entire image, part-based methods aim to learn based on accurate body parts (i.e., head, torso, arms, and legs). Optimal part-based models are (1) pose sensitive (i.e., for similar poses, shows strong activations); (2) extendable to all samples; (3) discriminative on extracting features. CNNs can handle all these factors to some extend, and [77] empirical experiments confirm that for deeper networks, accurate body-parts are less significant.

As one of the first part-based works, inspired by a part detector (i.e., deformable part model [78], which captures viewpoint and pose variations), Zhang et al. [79] propose two descriptors that learn based on the part annotations. Their main objective is to localize the semantic parts and obtain a normalized pose representation. To this end, the first descriptor is fed by correlated body parts, while for the second descriptor, the input body splits have no semantic correlation. Intuitively, the first

descriptor is based on the inherent semantics of the input image, and the second descriptor learns the cross-component correspondences between the body parts.

Later, in this context, Reference [77] proposes a model composed of a CNN-based body-part detector, including an SVM classifier (trained on the full-body and body parts, that is, head, torso, and legs) to predict the human attributes and action. Given an input image, a Gaussian pyramid is obtained, each level is fed to several convolutional layers to produce pyramids of feature maps. The convolution of each feature-level with each body-part produces scores correspond to that body-part. Therefore, the final output is a pyramid of part model scores suitable for learning an SVM classifier. The experiments indicate that using body-part analysis and making the network deeper improve the results.

As earlier part-based methods used separate feature extractors and classifiers, the parts could not be optimized for recognizing the semantic attributes. Moreover, the detectors, at that time, were inaccurate in detection. Therefore, Reference [80] proposed an end-to-end model, in which the body partitions are generated based on the skeleton information. As authors augment a large skeleton estimation dataset (MPII [81]) for human skeleton information (which is less prone to error for annotation in comparison with bounding box annotations for body parts), their body detector is more accurate in detecting the relevant partitions, leading to better performance.

To encode both global and fine-grained features and implicitly relate them to the specific attributes, References [82] proposes to add several branches on top of a ResNet50 network, such that each branch explores particular regions of the input data and learns an exclusive classifier. Meanwhile, before the classifier stage, all branches share a layer, which passes the 6 static regions of features to the attribute classifiers. For example, the head attribute classifier is fed only with the two upper strips of the feature maps. Experimental results on the Market-1501 dataset [24] show that applying a layer that feeds regional features to the related classifiers can improve the $mA$ from 85.0 to 86.2. Further, repeating the experiments while adding a branch to the architecture of the model for predicting the person ID (as an extra-label) improves the $mA$ result from 84.9 to 86.1. These experiments show that simultaneous ID prediction without any purpose could slightly diminish the accuracy.

### 3.1.4. Attention Based Methods

By focusing on the most relevant regions of the input data, human beings recognize the objects and their attributes without the background's interference. For example, when recognizing the head-accessories attributes of an individual, special attention is given to the facial region. Therefore, many HAR methods have attempted to implement an attention module to be inserted at multiple levels of CNN. Attention heat maps (also called localization score map [83,84] or class activation map [85]) are colorful localization score maps that make the model interpretable and are usually faded over the original image to show the model's ability to focus on the relevant regions.

In order to eliminate the need for body-part detection and prior correspondence among the patches, Reference [86] proposed to refine the Class Activation Map network [85], in which the relevant regions of the image to each attribute are highlighted. The model comprises a CNN feature extraction backbone with several branches on its top, which yield the scores for all the attributes and their regional heat maps. The fitness of the attention heat maps is measured using an exponential loss function, while the score of the attributes is derived from a classification loss function. The evaluation of the model is performed using two different convolutional backbones (i.e., VGG16 [87] and AlexNet [13]), and the result for the deeper network (VGG16) is better than the other one.

To extract more distinctive global and local features, Liu et al. [88] propose an attention module that fuses several feature layers of the relevant regions and yields attention maps. To take full advantage of the attention mechanism, they apply the attention module to different model levels. Obtaining the attentive feature maps from various layers of the network means that the model has captured multiple levels of the input sample's visual patterns so that the attention maps from higher blocks can cover more extensive regions, and the lower blocks focus on smaller regions of the input data.

Considering the problem of cloth classification and landmark detection, Reference [89] proposes an attentive fashion grammar network, in which both the symmetry of the cloths and effect of body motion is captured. To enhance the clothing classification, authors suggest to (1) develop supervised attention using the ground truth landmarks to learn the functional parts of the clothes and (2) use a bottom-up, top-down network [90], in which a successive down and up-sampling are performed on the attention maps to learn the global attention. The evaluation results of their model for clothing attribute prediction improved the counterpart methods by a large margin (30% to 60% top-5 accuracy on the DeepFashoin-C dataset [91]).

With a view to select the discriminative regions of the input data, Reference [92] proposes a model considering three aspects: (1) Using the parsing technique [93], they split features of each body-part and help the model learns the location-oriented features by pixel-to-pixel supervision. (2) Multiple attention maps are assigned to each label due to empowering the features from the relevant regions to that label and suppressing the other features. Different from the previous step, the supervision in this module is performed on the image-level. (3) Another module learns the relevant regions for all the attributes and learns from a global perspective. The quantitative results on several datasets show that the full version of the model improves the plain model's performance slightly (e.g., for the RAP dataset, the F1 metric improves from 79.15 to 79.98).

Reference [94] is another research that has focused on localizing the human attributes engaging multi-level attention mechanisms in full-frame images. First, supervised coarse learning is performed on the target person, in which the extracted features of each residual block is multiplied by the ground truth mask. Then, inspired by Reference [95], to further boost the attribute-based localization, an attention module uses the labels to refine the aggregated features of multiple levels of the model.

To alleviate the complex background and occlusion challenges in HAR, Reference [96] introduces a coarse attention layer that uses the multiplication between the output of the CNN backbone and ground truth human masks. Further, to guide the model to consider the semantic relationships among the attributes, authors use a multi-task architecture with a weighted loss function. This way, the CNN learns to find the relevant regions to the attributes in the foreground regions. Their ablation studies show that considering the correlation between attributes (multi-task learning) is more effective than coarse attention on the foreground region, although both improve the model performance.

### 3.1.5. Attribute Based Methods

Noticing the effectiveness of the additional information (e.g., pose, body-part and viewpoint) in the global feature representation, Reference [97] introduces a method that improves the localization ability of the model by locating the attributes' regions in the images. The model comprises two branches, one of them extracts the global features and provides the CAMs [98] (attention heat-maps), and the other one uses [99] to produce some regions of interests (RoI) for extracting the local features. To localize each attribute, the authors consider regions with high overlap between the CAMs and RoIs as the attribute location. Finally, the local and global features are aggregated using an element-wise sum. Their ablation studies on the RAP dataset show that for the model without localization $F1$ metric is about 77%, while the full-version model improves the results to about 80%.

As a weakly supervised method, Reference [100] aims to learn the regions in the input data related to the specific attributes. Thereby, the input image is fed into a BN-Inception model [101], and the features from three levels of the model (low, mid, and high) are concatenated together to be ready for three separate localization process. The localization module is built from a SE-Net [102] (that considers the channel relationships) proceeded with a Spatial Transformer Network (STN) (that performs conditional transformations on the feature maps) [65]. The training is weakly supervised because instead of using the ground truth coordinates of the attribute region, the STN is treated as a differentiable RoI pooling layer that is learned without box annotations. The $F1$ metric on the RAP dataset for BN-Inception plain model is around 78.2 while this number fro the full version of the model is 80.2.

Considering that both the local and global features are important for making a prediction, most of the literature's localization-based methods have introduced modular techniques. Therefore, the proposed module could be used in multiple levels of the model (from the first convolutional layers to the final classification layers) to capture both the low-level and high-level features. Intuitively, the implicit location of each attribute is learned in a weakly supervised manner.

### 3.2. Limited Data

Although deep neural networks are powerful in the attribute recognition task, an insufficient amount of data causes an early overfitting problem and hinders them from extracting a generalized feature representation from the input data. Meanwhile, the deeper the networks are, the more data are required to learn a wide range of layer weight parameters. Data augmentation and transfer learning are two primary solutions that address the challenge of limited data in computer vision tasks. In the context of HAR, there are few researches that have studied the effectiveness of these methods that are discussed in the following.

(A) Data Augmentatio. In this context, Bekele et al. [103] studied the effectiveness of 3 basic data augmentation techniques on their proposed solution and observed that the *F*1 score is improved from 85.7 to 86.4 for an experiment on the PETA dataset. Further, [104] discussed that ResNets could take advantage of the skipped connections to avoid overfitting. Their experimental results on the PETA dataset confirm the superiority of ResNets without augmentation over the SVM-based and plain CNN models.

(B) Transfer Learning. In clothing attribute recognition, some works may deal with two domains (types of images): (1) in shop images that are high-quality in specific poses; (2) in-the-wild images that vary in the pose, illumination, and resolution. To address the problem of limited labeled data, we can transfer the knowledge of one domain to the other domain. In this context, inspired by curriculum learning, Dong et al. [105] suggest a two-step framework for curriculum transfer of knowledge from shop clothing images to in-the-wild *similar* clothing images. To this end, they train a multi-task network with easy samples (in-shop) and copy its weights to a triplet-branch curriculum transfer network. At first, these branches have identical weights; however, in the second training stage (with harder examples), the feature similarity values between the target and the positive branches become larger than between the target and negative branches. The ablation studies confirm the effectiveness of the authors' idea and show that the mean average ($mA$) improved from 51.4 to 58.8 for plain multi-task and proposed model, respectively, on the Cross-Domain clothing dataset [106]. Moreover, this work indicates that curriculum learning versus end-to-end learning achieves better results, with 62.3 and 64.4 of $mA$, respectively.

### 3.3. Attributes Relationship

Both the spatial and semantic relationships among the attributes affect the performance of the PAR models. For example, hairstyle and footwear are correlated, while related to different regions (i.e., spatial distributions) of the input data. Regarding the semantic relationship, pedestrian attributes may either conflict with each other or are mutually confirming. For instance, wearing jeans and a skirt is an unexpected outfit, while wearing a T-shirt and sports shoes may co-appear with high probability. Therefore, taking these intuitive interpretations into account could be considered as a refinement step that improves the prediction-list of the attributes [107]. Furthermore, considering the contextual relation between various regions improve the performance of the PAR models. To consider the correlation among the attributes there are several possibilities such as using multi-task architecture [96], multi-label classification with weighted loss function [108], Recurrent Neural Networks (RNN) [109], Graph Convolutional Network (GCN) [110]. We have classified them into two main groups:

- Network-Oriented methods that take advantage of the various implementation of convolutional layers/blocks to discover the relation between attributes,

- math-oriented methods that may or may not extract the features using CNNs, but perform some mathematical operations on the features to modify them regarding the existing intrinsic correlations among the attributes.

In the following, we discuss the literature of both categories.

### 3.3.1. Network-Oriented Attribute Correlation Consideration

(A) Multi-task Learning. In [55], Lu et al. discuss that the intuition-based design of multi-task models is not an optimal solution for sharing the relevant information over multiple tasks, and they propose to gradually widen the structure of the model (add new branches) using an iterative algorithm. Consequently, in the final architecture, correlated tasks share most of the convolutional blocks together, while uncorrelated tasks will use different branches. Evaluation of the model on the fashion dataset [91] shows that by widening the network to 32 branches, the accuracy of the model cannot compete with other counterparts; however, the speed increases (from 34 ms to 10 ms) and the number of parameters decreases from 134 million to 10.5 million.

In a multi-task attribute recognition problem, each task may have a different convergence rate. To alleviate this problem and jointly learn multiple tasks, Reference [111] proposes a weighted loss function that updates the weights for each task in the course of learning. The experimental evaluation on the Market-1501 dataset [24] shows an improvement in accuracy from 86.8% to 88.5%.

In [112,113], the authors study the multi-task nature of PAR and attempt to build an optimal grouping of the correlated tasks, based on which they share the knowledge between tasks. The intuition is that, similar to the human brain, the model should learn more manageable tasks first and then uses them for solving more complex tasks. The authors claim that learning correlated tasks needs less effort, while uncorrelated tasks require specific feature representations. Therefore, they apply a curriculum learning schedule to transfer the knowledge of the easier tasks (strongly correlated) to the harder ones (weakly correlated). The baseline results show that learning the tasks individually yields 71.0% accuracy on the SoBiR dataset [114], while this number for learning multiple tasks at once is 71.3% and for a curriculum-based multi-task model is 74.2%.

Considering HAR as a multi-task problem, Reference [54] proposes to improve the model architecture in terms of feature sharing between tasks. Authors claim that by learning a linear combination of features, the inter-dependency of the channels is ignored, and the model cannot exchange spatial information. Therefore, after each convolutional block in the model, they insert a shared module between tasks to share the information. This module considers three aspects: (1) fusing the features of each two tasks together, (2) generating attention maps regarding the location of the attributes [115], and (3) keeping the effect of the original features of each task. Ablation studies over this module's positioning indicate that adding it at the end of the convolutional blocks yields the best results. However, the performance is approximately stable when different branches of the module (one at a time) are ablated.

(B) RNN. In [116], authors discuss that person re-id focuses on the global features, while attribute recognition relies on local aspects of individuals. Therefore, Liu et al. [116] propose a network consisted of three parts that work together to learn the person's attributes and re-identification (re-id). Further, to capture the contextual spatial relationships and focus to the location of each attribute, they use the RNN-CNN backbone feature extractor followed by an attention model.

To mine the relation of attributes, Reference [117] uses a model based on Long Short Term Memory (LSTM). Intuitively, using several successive stages of LSTM preserves the necessary information along the pipeline and forgets the uncorrelated features. In this work, the authors first detect three-body pose-lets based on the skeleton information. They consider the full-body as another pose-let followed by several fully connected layers to produce several groups of features (for each attribute, one group of features). Each group of features is passed to an LSTM block, followed by a fully-connected layer. Finally, the concatenation of all features is considered as the final feature representation of the input image. Considering that LSTM blocks are successively connected to each other, they carry the useful

information of previous groups of features to the next LSTM. The ablation study in this work shows that the plain Inception-v3 on PETA dataset attains 85.7 of *F*1 metric, and adding LSTM blocks on top of the baseline improves its performance to 86.0, while the full version of the model that processes the body-parts achieves to *F*1 86.5.

Regarding the functionality of RNN in contextual combinations in the sequenced data, Reference [118] introduces two different methods to localize the semantic attributes and capture their correlations implicitly. In the first method, the input image's extracted features are divided into several groups; then, each group of features is given to an LSTM layer followed by a regular convolution block and a fully connected layer, while all the LSTM layers are connected together successively. In the second method, all the extracted features from the backbone are multiplied (spatial point-wise multiplication) by the last convolution block's output to provide global attention. The experiments show that dividing the features into groups from global to local features yields better results than random selection.

Inspired by image-captioning methods, Reference [119] introduced a Neural PAR that converts attributes recognition to the image-captioning task. To this end, they generated sentence vectors to describe each pedestrian image using a random combination of attribute-words. However, there are two major disruptions in designing an image-caption architecture for attribute classification: (1) variable length of sentences (attribute-words) for different pedestrians and (2) finding relevance between attributes vectors and spatial space. To address these challenges, the authors used RNNs units and lookup-table, respectively. how much they improved the results in comparison with a plain network? how they implemented this idea?

To deal with low-resolution images, Wang et al. [109] formulated the PAR task as a sequential prediction problem, in which a two-step model is used to encode and decode the attributes for discovering both the context of intra-individual attributes and the inter-attribute relation. To this end, Wang et al. took advantage of LSTMs in both encode and decode steps for different purposes, such that in the encoding step the context of the intra-person attributes is learned, while in the decoding step, LSTMs is utilized to learn the inter-attributes correlation and predict the attributes as a sequence prediction problem. how much they improved the results in comparison with a plain network?

(C) GCN. In Reference [110], Li et al. introduce a sequential-based model that relies on two graph convolutional networks, in which the semantic attributes are used as the nodes of the first graph, and patches of the input image are used as the nodes of the second graph. To discover the correlation between regions and semantic attributes, they embedded the output of the first graph as the extra inputs into the second graph and vise versa (the output of the second graph is embedded as the extra inputs into the first graph). To avoid a closed loop in the architecture, they defined two separate feed-forward branches, such that the first branch receives the image patches and presents the spatial context representation of them. This representation is then mapped into the semantic space to produce the features that capture the similarity between regions. The second branch input is semantic attributes that are processed using a graph network and mapped into spatial graphs to capture the semantic-aware features. The output of both branches is fused to let and end-to-end learning. The ablation studies show that in comparison with a plain ResNet50 network, the *F*1 results could improve by margins of 3.5 and 1.3 for the PETA and RAP datasets, respectively.

Inspired by Reference [110], in Reference [107], Li et al. present a GCN-based model to yield the human parsing alongside the human attributes. Therefore, a graph is built upon the image features so that each group of features corresponds to one node of the graph. Afterward, to capture the relationships among the groups of attributes, a graph convolution is performed. Finally, for each node, a classifier is learned to predict the attributes. To produce the human parsing results, they apply a residual block that uses both the original features and the output of the graph convolution in the previous branch. Based on the ablation study, a plain ResNet50 on the PETA dataset achieves a *F*1 score of 85.0, while a model based on body parts yields a *F*1 score of 84.4, and this number for the model equipped with the above-mentioned idea is 87.9.

Tan et al. [120] observed the close relationship between some of the human attributes and claimed that in multi-task architectures, the final loss function layer is the critical point of learning, which may not have sufficient influence for obtaining a comprehensive representation for explaining the attribute correlations. Moreover, the limitation in receptive fields of CNNs [121] hinders the model's ability to effectively learn the contextual relations in the data. Therefore, to capture the structural connections among attributes and contextual information, the authors use two Graph Convolutional Networks (GCN) [122]. However, as image data is not originally structured as graphs, they use the extracted attribute-specific features (each feature corresponds to one attribute) from a ResNet backbone to obtain the first graph. For the second graph, clusters of regions (pixels) in the input image are considered as the network nodes. The clusters are learned using the share ResNet backbone—with the previous graph). Finally, the outputs of both graph-based branches are averaged. As LSTM also considers the relationship between parts, authors have replaced their proposed GCNs with LSTMs in the model and observed a slight drop in the model's performance. The ablation strides on three pedestrian datasets show that the *F*1 metric performance of a vanilla model improves with a margin of 2.

Reference [123] recognized the clothing style by mixing extracted features from the body parts. They applied a graph-based model with Conditional Random Fields (CRFs) to explore the correlation between clothes attributes. Specifically, using the weighted sum of body-part features, they trained an SVM for each of the attributes and used CRF to learn the relationships between attributes. By training the CRF with output probability scores from SVM classifiers, the attributes' relationship is explored. Although using CRFs was successful in this work, there are yet some disadvantages: (a) due to extensive computational cost, CRFs is not an appropriate solution when a broad set of attributes are considered, and (b) CRFs cannot capture the spatial relation between attributes [110] (c) models can not simultaneously optimize classifiers and CRFs [110], so it is not useful in an end-to-end model.

### 3.3.2. Math-Oriented Attribute Correlation Consideration

(A) Grammar. In [124], Park et al. addressed the need for an interpretable model that can jointly yield the body-pose information (body joints coordinates) and human semantic attributes. To this end, authors implemented an and-or grammar model, in which they integrated three types of grammars—(1) simple grammars that break down the full-body into smaller nodes; (2) dependency grammar that indicates which nodes (body parts) are connected to each other and models the geometric articulations; (3) attribute grammar that assigns the attributes to each node. The ablation studies for attribute prediction showed that the performance is better if the best pose estimation for each attribute is used for predicting the corresponding attribute score.

(B) Multiplication. In [125], authors discussed that a plain CNN could not handle human multi-attribute classifications effectively, as for each image, several labels have been entangled. To address this challenge, Han et al. [125] proposed to use a ResNet50 backbone followed by multiple branches to predict the occurrence probability of each attribute. Further, to improve the results, they provided a matrix from ground truth labels to obtain the conditional probability of each label (semantic attribute) given another attribute. The multiplication of this matrix by the previously obtained probability provides the models with a priori knowledge about the correlation of attributes. The ablation study indicated that the baseline (plain ResNet50) on the PETA dataset achieves 85.8 of *F*1 metric, while this number for a simple multi-branch model and full-version model is 86.6 and 87.6, respectively.

In order to mitigate the correlation between the visual appearance and the semantic attributes, Reference [126] uses a fusion attention mechanism and provides a balanced-weight between the image-guided and attribute-guided features. First, attributes are embedded in a latent space with the same dimension of the image features. Next, a nonlinear function is applied to the image features to obtain its feature distribution. Then, the image-guided features are obtained via an element-wise multiplication between the feature distribution of the image and the embedded attribute features. To obtain the attribute-guided features, they embed the attributes to a new latent space; next, the

results of the element-wise multiplication between image features and embedded attribute features are considered as the input of a nonlinear function, for which its output provides attribute-guided features. Meanwhile, to consider the class imbalance, authors use the focal loss function to train the model. The ablation study shows that the *F*1 metric performance of the baseline on the PETA dataset is 85.6, which improves to 85.9 when the model is equipped with the above-mentioned idea.

In Reference [127], authors propose a multi-task architecture, in which each attribute corresponds to one separate task. However, to consider the relationship between attributes, both the input image and category information are projected into another space, where the latent factors are disentangled. By applying the element-wise multiplication between the feature representation of the image and its class information, the authors define a discriminant function. When using it, a logistic regression model can learn all the attributes simultaneously. To show the efficiency of the methods, authors evaluate their proposed approach in several attribute datasets of animals, objects, and birds.

(C) Loss Function. Li et al. [128] discussed the attribute relationships and introduced two models to demonstrate the effectiveness of their idea. Considering HAR as a binary classification problem, the authors proposed a plain multi-label CNN that predicts all the attributes at-once. They also equipped the previous model with a weighted-loss function (cross-entropy), in which each attribute classifier has a specific weight to update the network weights for the next epoch. The experimental results on the PETA dataset with 35 attributes indicated that weighted cross-entropy loss function could improve the accuracy prediction in 28 attributes and increase the *mA* by 1.3 percent.

## 3.4. Occlusion

In HAR, occlusion is a primary challenge, in which parts of the useful information of the input data may be covered with other subjects/objects [129]. As this situation is likely to occur in real-world scenarios, it is necessary to be handled. In the context of person re-id, Reference [130] claims that inferring the occluded body parts could improve the results, and in the HAR context, Reference [131] suggests that using sequences of pedestrian images somehow alleviates the occlusion problem.

Considering the low-resolution images and partial occlusion of the pedestrian's body, Reference [132] proposed to manipulate the dataset with occurring frequent partial occlusions and degraded the resolution of the data. Then, the authors trained a model to rebuild the images with high resolution and do not suffer from occlusion. This way, the reconstruction model will help to manipulate the original dataset before training a classification model. As rebuild is performed with a GAN, the generated images are different from the original annotated dataset and somehow lost part of the annotations, which degrade the overall performance of the system compared to when one uses the original dataset for training. However, the ablation study in this paper shows that if two identical classification networks are separately trained on corrupted and generated data, the performance of the model that learns from the reconstructed data is better with a high margin.

To tackle the problem of occlusion, Reference [133] proposes to use a sequence of frames for recognizing human attributes. First, they extract the frame-level spatial features using a shared ResNet-50 backbone feature extractor [134]. The extracted features are then processed in two separate paths, one of them learns the body pose and motion, and the other branch learns the semantic attributes. Finally, each attribute's classifier uses an attention module that generates an attention vector showing the importance of each frame for attribute recognition.

To address the challenge of partial occlusion, References [129,131] adopted video datasets for attributes recognition as often occlusions are a temporary situation. Reference [129] divided each video clip to several pieces and extracted a random frame from each piece to create a new video clip with a few frame length. The final recognition confidence of each attribute is obtained by aggregating the recognition probability on the selected frames.

### 3.5. Classes Imbalance

The existence of large differences between the number of samples for each attribute (class) is known as data class imbalance. Generally, in multi-class classification problems, the ideal scenario would be to use the same amount of data for each class, in order to preserve the learning importance of all the classes at the same level. However, the classes in HAR datasets are naturally imbalanced since the number of samples of some attributes (e.g., wearing skirts) are lower than others (e.g., wearing jeans). Large class imbalance causes over-fitting in classes with limited data, while classes with large number of samples need more training epochs to converge. To address this challenge, some methods attempt to balance the number of samples in each class as a pre-processing step [135–137], which are called *hard solutions*. Hard solutions are classified into three groups—(1) up-sampling the minority classes, (2) down-sampling the large classes, and (3) generating new samples. On the other hand, *soft solutions* are interested in handling the data class imbalance by introducing new training methods [138] or novel loss functions, in which the importance of each class is weighted based on the frequencies of the data [139–141]. Furthermore, the combination of both solutions has been the subject of some studies [142].

### 3.5.1. Hard Solutions

The earlier hard solutions are focused either on interpolation between the samples [137,143], or clustering the dataset and oversampling by cluster-based methods [144]. The primary way of up-sampling in deep learning is to augment the existing samples –as discussed in Section 3.2. However, excessive up-sampling may lead to over-fitting when the classes are highly imbalanced. Therefore, some works down-sample the majority classes [145]. Random down-sampling may be an easy choice, but Reference [146] proposes to use the boundaries among the classes to remove redundant samples. However, loss of information is an inevitable part of down-sampling, as some samples are removed, which may carry useful information.

To address these problems, Fukui et al. [28] designed a multi-task CNN, in which classes (attributes) with fewer samples are given more importance in the learning phase. The batch of samples in conventional learning methods are selected randomly; therefore, the rare examples are less likely to be in the mini-batch. Meanwhile, data augmentation cannot be sufficient for balancing the dataset as ordinary data augmentation techniques generate new samples regardless of their rarity. Therefore, Fukui et al. [28] defines a rarity rate for each sample in the dataset and perform the augmentation for rare samples. Later, from the created mini-batches, those with appropriate sample balance are selected for training the model. The experimental results on a dataset with four attributes show a slight improvement in the average recognition rate, though the superiority is not consistent for all the attributes.

### 3.5.2. Soft Solutions

As previously mentioned, soft solutions focus on boosting the learning methods' performance, rather than merely increasing/decreasing the number of samples. Designing loss functions is a popular approach for guiding the model to take full advantage of the minority samples. For instance, Reference [126] proposes the combination of focal loss [147] and cross-entropy loss functions to introduce a focal cross-entropy loss function (see Section 3.3.2 for the analytical review over [126]).

Considering the success of curriculum learning [148] in other fields of studies, in Reference [138], the author addressed the challenge of imbalance-distributed data in HAR by batch-based adjustment of data sampling strategy and loss weights. It was argued that providing balanced distribution from a highly imbalanced dataset (using sampling strategies) for the whole learning process may cause the model to disregard the samples with most variations (i.e., classes with majority samples) and only emphasizes on the minority class. Moreover, the weighted terms in loss functions play an essential role in the learning process. Therefore, both the classification loss (often cross-entropy) and metric

learning loss (which aims to learn feature embedding for distinguishing between samples) should be handled based on their importance. To consider these aspects, authors defined two schedules, one for adjusting the sampling strategy by re-ordering the data from imbalanced to balanced and easy to hard; and the other curriculum schedule handles the loss importance between classification and distance metric learning. The ablation study in this work showed that the sampling scheduler could increase the results of a baseline model form 81.17 to 86.58, and adding loss scheduler to it could improve the results to 89.05.

To handle the class imbalance problem, Reference [149] modifies the focal loss function [147] and apply it for an attention-based model to focus on the hard samples. The main idea is to add a scaling factor to the binary cross-entropy loss function to down-weight the effect of easy samples with high confidence. Therefore, the hard misclassified samples of each attribute (class) add larger values to the loss function and become more critical. Considering the usual weakness of attention mechanism that does not consider the location of an attribute, the authors modified the attention masks in multiple levels of the model using attribute confidence weighting. Their ablation studies on the WIDER dataset [75] with ResNet-101 backbone feature extractor [134] showed the plain model achieves mA 83.7 and applying the weighted focal loss function improve the results to 84.4 while adding the multi-scale attention increased it to 85.9.

### 3.5.3. Hybrid Solutions

Hybrid approaches use the combination of the above-mentioned techniques. Performing data augmentation over the minority classes and applying a weighted loss function or a curriculum learning strategy are examples of hybrid solutions for handling the class data imbalance. In Reference [142], the authors discuss that learning from an unbalanced dataset leads to biased classification, with higher classification accuracy over the majority classes and lower performance over the minority classes. To address this issue, Chawla et al. [142] proposed an algorithm that focuses on difficult samples (misclassified). To implement this strategy, the authors took advantage of Reference [143], which generates new synthetic instances in each training iteration from the minority classes. Consequently, the weights for the minority samples (false negatives) are increased, which improves the model's performance.

### 3.6. Part-Based and Attribute Correlation-Based Methods

"Whether considering a group of attributes together improve the results of an attribute recognition model or not?" is the question that Reference [150] tries to answer by addressing the correlation between attributes using a Conditional Random Field (CRF) strategy. Concerning the calculated probability distribution over each attribute, all the Maximum A Posterioris (MAPs) are estimated, and then, the model searches for the most probable mixture in the input image. To also consider the location of each attribute, authors extract the part patches based on the bounding box around the full-body, as in fashion datasets pose variations are not significant. A comparison between several simple baselines shows that the CRF-based method ($0.516 F1$ score) works slightly better than a localization-based CNN ($0.512 F1$ score) on the Chictopia dataset [151], while a global-based CNN $F1$ performance is 0.464.

## 4. Datasets

As opposed to other surveys, instead of merely enumerating the datasets, in this manuscript, we discuss the advantages and drawbacks of each dataset, with emphasis on data collection methods/software. Finally, we discuss the intrinsically imbalanced nature of HAR datasets and other challenges that arise when gathering data.

*4.1. PAR Datasets*

- PETA dataset. **PE**des**T**rian **A**ttribute (PETA) [152] dataset combines 19,000 pedestrian images gathered from 10 publicly available datasets; therefore the images present large variations in terms of scene, lighting conditions and image resolution. The resolution of the images varies from 17 × 39 to 169 × 365 pixels. The dataset provides rich annotations: the images are manually labeled with 61 binary and 4 multi-class attributes. The binary attributes include information about demographics (gender: *Male*, age: *Age16–30, Age31–45,Age46–60, AgeAbove61*), appearance (*long hair*), clothing (*T-shirt, Trousers* etc.) and accessories (*Sunglasses, Hat, Backpack* etc.). The multi-class attributes are related to (eleven basic) color(s) for the upper-body and lower-body clothing, shoe-wear, and hair of the subject. When gathering the dataset, the authors tried to balance the binary attributes; in their convention, a binary class is considered balanced if the maximal and minimal class ratio is less than 20:1. In the final version of the dataset, more than half of the binary attributes (31 attributes) have a balanced distribution.

- RAP dataset. Currently, there are two versions of the RAP (**R**ichly **A**nnotated **P**edestrian) dataset. The first version, RAP-v1 v1 [153] was collected from a surveillance camera in shopping malls over a period of three months; next, 17 hours of video footage were manually selected for attribute annotation. In total, the dataset comprises 41,585 annotated human silhouettes. The 72 attributes labeled in this dataset include demographic information (*gender* and *age*), accessories (*backpack, single shoulder bag, handbag, plastic bag, paper bag* etc.), human appearance (*hair style, hair color, body shape*) and clothing information (*clothes style, clothes color, footware style, footware color* etc.). In addition, the dataset provides annotations about occlusions, viewpoints and body-parts information.

   The second version of the RAP dataset [108] is intended as a unifying benchmark for both person retrieval and person attribute recognition in real-world surveillance scenarios. The dataset was captured indoor, in a shopping mall and contains 84,928 images (2589 person identities) from 25 different scenes. High-resolution cameras (1280 × 720) were used to gather the dataset, and the resolution of human silhouettes varies from 33 × 81 to 415 × 583 pixels. The attributes annotated are the same as in RAP v2 (72 attributes, and occlusion, viewpoint, and body-parts information).

- DukeMTMC dataset. **Duke**MTMC-reid (**M**ulti-**T**arget, **M**ulti-**C**amera) dataset [154] was collected in Duke's university campus and contains more than 14 h of video sequences gathered from 8 cameras, positioned such that they capture crowded scenes. The main purpose of this dataset was person re-identification and multi-camera tracking; however, a subset of this dataset was annotated with human attributes. The annotations were provided at the identity level, and they included 23 attributes, regarding the gender (male, female), accessories: wearing hat (yes, no), carrying a backpack (yes, no), carrying a handbag (yes, no), carrying other types of the bag (yes, no), and clothing style: shoe type (boots, other shoes), the color of shoes (dark, bright), length of upper-body clothing (long, short), 8 colors of upper-body clothing (black, white, red, purple, gray, blue, green, brown) and 7 colors of lower-body clothing (black, white, red, gray, blue, green, brown). Due to violation of civil and human rights, as well as privacy issues, since June 2019, Duke University has terminated the DukeMTMC dataset page.

- PA-100K dataset. PA-100k dataset [88] was developed with the intention to surpass the existing HAR datasets both in quantity and in diversity; the dataset contains more than 100,000 images captured in 598 different scenarios. The dataset was captured by outdoor surveillance cameras; therefore, the images provide large variance in image resolution, lighting conditions, and environment. The dataset is annotated with 26 attributes, including demographic (age, gender), accessories (handbag, phone) and clothing information.

- Market-1501 dataset. Market-1501 attribute [24,155] dataset is a version of the Market-1501 dataset augmented with the annotation of 27 attributes. Market-1501 was initially intended for cross camera person re-identification, and it was collected outdoor in front of a supermarket using 6

cameras (5 high-resolution cameras and one low resolution). The attributes are provided at the identity level, and in total, there are 1501 annotated identities. In total, the dataset has 32,668 bounding boxes for these 1501 identities. The attributes annotated in *Market-1501 attribute* include demographic information (gender and age), information about accessories (*wearing hat, carrying backpack, carrying bag, carrying handbag*), appearance (*hair length*) and clothing type and color (*sleeve length, length of lower-body clothing, type of lower-body clothing, 8 color of upper-body clothing, 9 color of lower-body clothing*).

- P-DESTRE Dataset. Over the recent years, as their cost has diminished considerably, UAVs applications extended rapidly in various surveillance scenarios. As a response, several UAVs datasets have been collected and made publicly available to the scientific community. Most of them are intended for human detection [156,157], action recognition [158] or re-identification [159]. To the best of our knowledge, the P-DESTRE [160] dataset is the first benchmark that addresses the problem of HAR from aerial images.

  P-DESTRE dataset [160] was collected in the campuses of two Universities from India and Portugal, using DJI-Phantom 4 drones controlled by human operators. The dataset provides annotations both for person re-identification, as well as for attribute recognition. The identities are consistent across multiple days. The annotations for the attributes include demographic information: *gender, ethnicity* and *age*, appearance information: *height, body volume, hair color, hairstyle, beard, moustache*; accessories information: *glasses, head accessories, body accessories*; *clothing* information and *action* information. In total, the dataset contains over 14 million person bounding boxes, belonging to 261 known identities.

### 4.2. FAR Datasets

- Parse27k dataset. **P**edestrian **a**ttribute **r**ecognition in **se**quences (Parse27k) dataset [161] contains over 27,000 pedestrian images, annotated with 10 attributes. The images were captured by a moving camera across a city environment; every 15th video frame was fed to the Deformable Part Model(DPM) pedestrian detector [78] and the resulting bounding boxes were annotated with the 10 attributes based on binary or multinomial propositions. As opposed to other datasets, the authors also included an N/A state (i.e., the labeler cannot decide on that attribute). The attributes from this dataset include gender information (3 categories—*male, female, N/A*), accessories (*Bag on Left Shoulder, Bag on Right Shoulder Bag in Left Hand, Bag in Right Hand, Backpack*; each with three possible states: *yes, no, N/A*), orientation (with *4 + N/A* or *8 + N/A* discretizations) and action attributes: *posture* (*standing, walking, sitting* and *N/A*) and *isPushing* (*yes, no, N/A*). As the images were initially processed by a pedestrian detector, the images of this dataset consist of a fixed-size bounding region of interest, and thus are strongly aligned and contain only a subset of possible human poses.

- CRP dataset. CRP (**C**altech **R**oadside **P**edestrians) [162] dataset was captured in real world conditions, from a moving vehicle. The position (bounding-box) of each pedestrian, together with 14 body joints are annotated in each video frame. CRP comprises 4222 video tracks, with 27,454 pedestrian bounding boxes. The following attributes are annotated for each pedestrian—age (5 categories: *child, teen, young adult, middle aged* and *senior*), gender (2 categories—*female* and *male*), weight (3 categories: *Under, Healthy* and *Over*), and clothing style (4 categories—*casual, light athletic, workout* and *dressy*). The original, un-cropped videos together with the annotations are publicly available.

- Describing People dataset. Describing People dataset [68] comprises 8035 images from the H3D [163] and the PASCAL VOC 2010 [164] datasets. The images from this database are aligned, in the sense that for each person, the image is cropped (by leaving some margin) and then scaled so that the distance between the hips and the shoulders is 200 pixels. The dataset features 9 binary (True/False) attributes, as follows: gender (*is male*), appearance (*long hair*), accessories (*glasses*)

and several clothing attributes (*has hat, has t-shirt, has shorts, has jeans, long sleeves, long pants*). The dataset was annotated on Amazon Mechanical Turk by five independent labelers; the authors considered a valid label if at least four of the five annotators agreed on its value.

- HAT dataset. **H**uman **AT**tributes (HAT) [66,78] contains 9344 images gathered from Flickr; for this purpose, the authors used more than 320 manually specified queries to retrieve images related to people and then, employed an off-the-shelf person detector to crop the humans in the images. The false positives were manually removed. Next, the images were labeled with 27 binary attributes; these attributes incorporate information about the gender (Female), age (Small baby, Small kid, Teen aged, Young (college), Middle Aged, Elderly), clothing (Wearing tank top, Wearing tee shirt, Wearing casual jacket, Formal men suit, Female long skirt, Female short skirt, Wearing short shorts, Low cut top, Female in swim suit, Female wedding dress, Bermuda/beach shorts), pose (Frontal pose, Side pose, Turned Back), Action (Standing Straight, Sitting, Running/Walking, Crouching/bent, Arms bent/crossed) and occlusions (Upper body). The images have high variations both in image size and in the subject's position.

- WIDER dataset. WIDER Attribute dataset [75] comprises a subset of 13,789 images selected from the WIDER database [165], by discarding the images full of non-human objects and the images in which the human attributes are indistinguishable; the human bounding boxes from these images are annotated with 14 attributes. The images contain multiple humans under different and complex variations. For each image, the authors selected a maximum of 20 bounding boxes (based on their resolution), so in total, there are more than 57,524 annotated individuals. The attributes follow a ternary taxonomy: positive, negative and unspecified, and include information about age (*Male*), clothing (*Tshirt, longSleeve, Formal, Shorts, Jeans, Long Pants, Skirt*), *accessories (Sunglasses, Hat, Face Mask, Logo)*, appearance (*Long Hair*). In addition, each image is annotated into one of 30 event classes (meeting, picnic, parade, etc.), thus allowing to correlate the human attributes with the context they were perceived in.

- CAD dataset. **C**lothing **A**ttributes **D**ataset [123] uses images gathered from the website Sartorialist (https://www.thesartorialist.com/) and Flikcr. The authors downloaded several images, mostly of pedestrians, and applied an upper-body detector to detect humans; they ended up with 1856 images. Next, the ground truth was established by labelers from Amazon Mechanical Turk. Each image was annotated by 6 independent individuals, and a label was accepted as ground truth if it has at least 5 agreements. The dataset is annotated with the gender of the wearer, information about the accessories (*Wearing scarf, Collar presence, Placket presence*) and with several attributes regarding the clothing appearance (*clothing pattern, major color, clothing category, neckline shape* etc.).

- APiS dataset. The **A**ttributed **P**edestrians **in** **S**urveillance dataset [166] gathers images from four different sources: KITTI database [167], CBCL Street Scenes [168] (http://cbcl.mit.edu/software-datasets/streetscenes/), INRIA database [48] and some video sequences collected by the authors at a train station; in total APiS comprises 3661 images. The human bounding boxes are detected using an off-the-shelf pedestrian detector, and the results are manually processed by the authors: the false positives and the low-resolution images (smaller than 90 pixels in height and 35 pixels in width) are discarded. Finally, all the images of the dataset are normalized in the sense that the cropped pedestrian images are scaled to $128 \times 48$ pixels. These cropped images are annotated with 11 ternary attributes (positive, negative, and ambiguous) and 2 multi-class attributes. These annotations include demographic (gender) and appearance attributes (*long hair*), as well as information about accessories (*back bag, S-S (Single Shoulder) bag, hand carrying*) and clothing (*shirt, T-shirt, long pants, M-S (Medium and Short) pants, long jeans, skirt, upper-body clothing color, lower-body clothing color*). The multi-class attributes are the two attributes related to the clothing color. The annotation process is performed manually and divided into two stages: annotation stage (the independent labeling of each attribute) and validation stage (which exploits

the relationship between the attributes to check the annotation; also, in this stage, the controversial attributes are marked as ambiguous).

## 4.3. Fashion Datasets

- DeepFashion Dataset. The DeepFashion dataset [91] was gathered from shopping websites, as well as image search engines (blogs, forums, user-generated content). In the first stage, the authors downloaded 1,320,078 images from shopping websites and 1,273,150 images from Google images. After a data cleaning process, in which duplicate, out-of-scope, and low-quality images were removed, 800,000 clothing images were finally selected to be included in the DeepFashion dataset. The images are annotated solely with clothing information; these annotations are divided into categories (50 labels: dress, blouse, etc.) and attributes (1000 labels: adjectives describing the categories). The categories were annotated by expert labelers, while for the attributes, due to their huge number, the authors resorted to meta-data annotation (provided by Google search engine or by the shopping website). In addition, a set of clothing landmarks, as well as their visibility, are provided for each image.

  DeepFashion is split into several benchmarks for different purposes: category and attribute prediction (classification of the categories and the attributes), in-shop clothes retrieval (determine if two images belong to the same clothing item), consumer-to-shop clothes retrieval (matching consumer images to their shop counterparts) and fashion landmark detection.

**Table 1.** Pedestrian attributes datasets.

| Dataset Type | Dataset | #images | Demographic | Accessories | Appearance | Clothing | Colour | Setup |
|---|---|---|---|---|---|---|---|---|
| Pedestrian | PETA [152] | 19,000 | ✓ | ✓ | ✓ | ✓ | ✓ | 10 databases |
| | RAP v1 [153] | 41,585 | ✓ | ✓ | ✓ | ✓ | ✓ | indoor static camera |
| | RAP v2 [108] | 84,928 | ✓ | ✓ | ✓ | ✓ | ✓ | indoor static camera |
| | DukeMTMC † | 34,183 | ✓ | ✓ | ✗ | ✓ | ✓ | outdoor static camera |
| | PA-100K [88] | 100,000 | ✓ | ✓ | ✗ | ✓ | ✗ | outdoor, surveillance cameras |
| | Market-1501 [24] | 1501 | ✓ | ✓ | ✓ | ✓ | ✓ | outdoor |
| | P-DESTRE [160] | 14M | ✓ | ✓ | ✓ | ✓ | ✗ | UAV |
| Full body | Parse27k [161] | 27,000 | ✓ | ✓ | ✗ | ✗ | ✗ | outdoor moving camera |
| | CRP [162] | 27,454 | ✓ | ✓ | ✗ | ✗ | ✗ | moving vehicle |
| | APiS [166] | 3661 | ✓ | ✓ | ✓ | ✓ | ✓ | 3 databases |
| | HAT [66] | 9344 | ✓ | ✓ | ✗ | ✓ | ✗ | Flickr |
| | CAD [123] | 1856 | ✓ | ✓ | ✗ | ✓ | ✓ | website crawling |
| | Describing People [68] | 8035 | ✓ | ✓ | ✗ | ✓ | ✗ | 2 databases |
| | WIDER [75] | 13,789 | ✓ | ✓ | ✓ | ✓ | ✗ | website crawling |
| Synthetic | CTD [169] | 880 | ✗ | ✗ | ✗ | ✓ | ✓ | generated data |
| | CLOTH3D [170] | 2.1M | ✗ | ✗ | ✗ | ✓ | ✓ | generated data |

† Permanently suspended regarding privacy issues.

## 4.4. Synthetic Datasets

Virtual reality systems and synthetic image generation have become prevalent in the last few years, and their results are more and more realistic and of high resolution. Therefore, we also discuss some data sources comprising computer-generated images. It is a well-known fact that the performance of deep learning methods is highly dependent on the amount and distribution of data they were trained on, and synthetic datasets could theoretically be used as an inexhaustible source of diverse and balanced data. In theory, any combination of attributes in any amount could be synthetically generated.

- DeepFashion—Fashion Image Synthesis. The authors of DeepFashion [91] introduce FashionGAN, an adversarial network for generating clothing images on a wearer [171]. FashionGAN is organized into two stages: on a first level, the network generates a semantic segmentation map modeling the wearer's pose. In the second level, a generative model renders an image with

precise regions and textures conditioned on this map. In this context, the DeepFashion dataset was extended with 78,979 images (taken for the In-shop Clothes Benchmark), associated with several caption sentences and a segmentation map.

- CTD Dataset. Clothing Tightness dataset [169] (CTD) comprises 880 3D human models, under various poses, both static and dynamic, "dressed" with 228 different outfits. The garments in the dataset are grouped under various categories, such as "T/long shirt, short/long/down coat, hooded jacket, pants, and skirt/dress, ranging from ultra-tight to puffy". CTD was gathered in the context of a deep learning method that maps a 3D human scan into a hybrid geometry image. This synthetic dataset has important implications in virtual try-on systems, soft biometrics, and body pose evaluation. The main drawbacks of this dataset are that it cannot capture exaggerated human postures of low 3D human scans.

- CLOTH3D Dataset. CLOTH3D [170] comprises thousands of 3D sequences of animated human silhouettes, "dressed" with different garments. The dataset features a large variation on the garment shape, fabric, size, and tightness, as well as human pose. The main applications of this dataset listed by the authors include—"human pose and action recognition in-depth images, garment motion analysis, filling missing vertices of scanned bodies with additional metadata (e.g., garment segments), support designers and animators tasks, or estimating 3D garment from RGB images".

## 5. Evaluation Metrics

This section reviews the most common metrics used in the evaluation of HAR methods. Considering that HAR is a multi-class classification problem, Accuracy (*Acc*), Precision (*Prec*), Recall (*Rec*), and *F*1 score are the most common metrics for measuring the performance of these methods. In general, these metrics can be calculated at two different levels: label-level and sample-level.

The evaluation at label-level considers each attribute independently. As an example, if the gender and height attributes are considered with the labels (male, female) and (short, medium, high), respectively, the label-level evaluation will measure the performance of each attribute-label combination. The metric adopted in most papers for label-level evaluation is the mean accuracy (*mA*):

$$ mA = \frac{1}{2N} \sum_{i=1}^{N} \left( \frac{TP_i}{P_i} + \frac{TN_i}{N_i} \right), \tag{1} $$

where $i$ refers to each of the $N$ attributes. $mA$ determines the average accuracy between the positive and negative examples of each attribute.

In the sample-level evaluation, the performance is measured for each attribute disregarding the number of labels that it comprises. *Prec*, *Rec*, *Acc*, and *F*1 score for the $i^{th}$ attribute are thus given by:

$$ Prec_i = \frac{TP_i}{P_i}, \quad Rec_i = \frac{TP_i}{N_i}, \quad Acc_i = \frac{TP_i + TN_i}{P_i + N_i}, \quad F_i = \frac{2 * Prec * Rec}{Prec + Rec}. \tag{2} $$

The use of these metrics is very common for providing a comparative analysis of the different attributes. The overall system performance can be either measured by the mean $Acc_i$ over all the attributes or using $mA$. However, these metrics can diverge significantly, when attributes are highly unbalanced. $mA$ is preferred when authors deliberately want to evaluate the effect of data unbalancing.

## 6. Discussion

### 6.1. Discussion Over HAR Datasets

In recent years, HAR has received much interest from the scientific community, with a relatively large number of datasets developed for this purpose; this is also demonstrated by the number of citations. We performed a query for each HAR related database on the Google Scholar (scholar.google.

com) search engine, and extracted its corresponding number of citations; the results are graphically presented in Figure 3. In the past decade, more than 15 databases related to this research field have been published, and most of them received hundreds of citations.

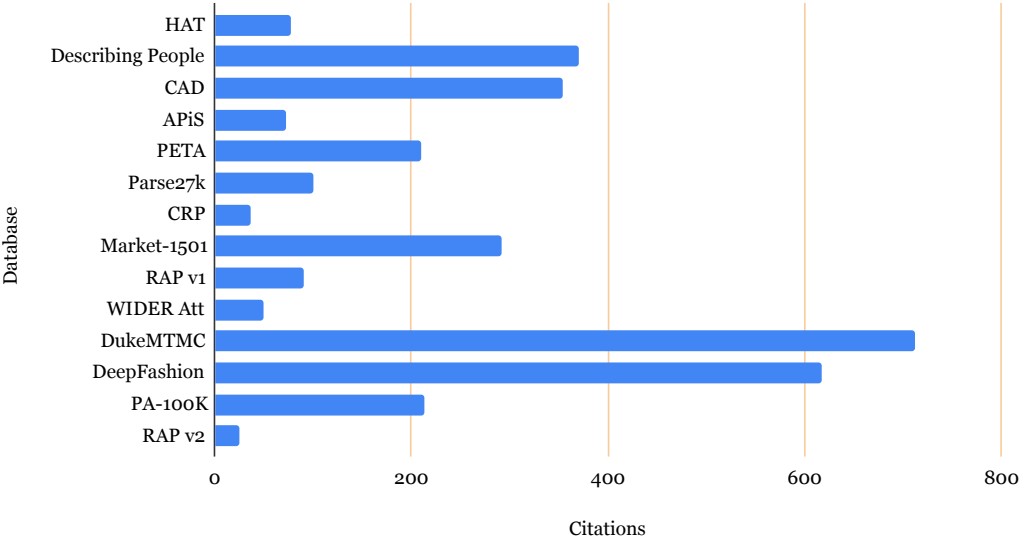

**Figure 3.** Number of citations to HAR datasets. The datasets are arranged in an increasing order by their publication date. The "oldest" dataset being HAT, published in 2009, while the latest is RAP v2, published in 2018.

In Table 1, we chose to taxonomize the attributes semantically into demographic attributes (gender, age, ethnicity), appearance attributes (related to the appearance of the subject, such as hairstyle, hair color, weight, etc.), accessory information (which indicate the presence of a certain accessory, such as a hat, handbag, backpack etc.) and clothing attributes (which describe the garments worn by the subjects). In total, we have described 17 datasets, the majority containing over ten thousand images. These datasets can be seen as a continuous effort made by researchers to provide large amounts of varied data required by the latest deep learning neural networks.

1. Attributes definition. The first issues that should be addressed when developing a new dataset for HAR are: (1) *which attributes should be annotated?* and (2) *how many and which classes are required to describe an attribute properly?*. Obviously, both these questions depend on the application domain of the HAR system. Generally, the ultimate goal on a HAR, regardless of the application domain, would be to accurately describe an image in terms of human-understandable semantic labels, for example, "a five-year-old boy, dressed in blue jeans, with a yellow T-shirt carrying a striped backpack". As for the second question, the answer is straightforward for some attributes, such as gender, but it becomes more complex and subjective for other attributes, such as age or clothing information. Let's take for example, the age label; different datasets provided different classes for this information: PETA distinguishes between *AgeLess15, Age16-30, Age31-45, Age46-60, AgeAbove61*, while CRP dataset adopted a different age classification scheme: *child*, *teen*, *young adult*, *middle aged* and *senior*. Now, if a HAR analyzer is integrated into a surveillance system in a crowded environment, such as Disneyland, and this system should be used to locate a missing child, the age labels from the PETA dataset are not detailed enough, as the "lowest" age class is AgeLess15. Secondly, these differences between the different taxonomies make it difficult to assess the performance of a newly developed algorithm across different datasets.

2. Unbalanced data. An important issue in any dataset is related to unbalanced data. Although some datasets were developed by explicitly striking for balanced classes, some classes are not that

frequent (especially those related to clothing information), and fully balanced datasets are not a trivial problem. The problem of imbalance also affects the demographic attributes. In all HAR datasets, the class of young children is poorly represented. To illustrate the problem of unbalanced classes, we selected two of the most prominent HAR related datatsets which are labeled with age information: CRP and PETA. In Figure 4, for each of these two datasets, we plot a pie charts to show age distribution of the labeled images.

Furthermore, as datasets are usually gathered in a single region (city, country, continent), the data tends to be unbalanced in terms of ethnicity. This is an important issue as some studies [172] proved the existence of *the other race effect* —the tendency to more easily recognize faces from the same ethnicity-– for machine learning classifier.

3. Data context. Strongly linked to the problem of data unbalance is the context or environment in which the frames were captured. The environment has a great influence on the distribution of the clothing and demographic (age, gender) attributes. In [75] the authors noticed "strong correlations between image event and the frequent human attributes in it". This is quite logical, as one would expect to encounter more casual outfits in a picnic or sporting event, while at ceremonies (wedding, graduation proms), people tend to be more elegant and dressed-up. The same is valid for the demographic attributes: if the frames are captured in the backyard of a kindergarten, one would that most of the subjects to be children. Ideally, a HAR dataset should provide images captured from multiple and variate scenes. Some datasets explicitly annotated the context in which the data was captured [75], while others address this issue by merging images from various datasets [152]. From another point of view, this leads our discussion to how the images from the datasets are presented. Generally speaking, the dataset provides the images either aligned (all the images have the same size and cropped around the human silhouette with a predefined margin; for example, [68]), or make the full video frame/image available and specify the bounding box of each human in the image. We consider that the latter approach is preferable, as it also incorporates context information and allows researches to decide how to handle the input data.

4. Binary attributes. Another question in database annotation is what happens when the attribute to annotate is indistinguishable due to low resolution and degraded images, occlusions, or other ambiguities. The majority of datasets tend to ignore this problem and classify the presence of an attribute or provide a multi-class attribute scheme. However, in a real-world setup, we cannot afford this luxury, as the case of indistinguishable attributes might occur quite frequently. Therefore, some datasets [161,166] formulate the attribute classification task with N + 1 classes (+1 for the N/A label). This approach is preferable, as it allows taking both views over the data: depending on the application context, one could simply ignore the N/A attributes or, make the classification problem more interesting, integrate the N/A value into the classification framework.

5. **Camera configuration.** Another aspect that should be taken into account when discussing HAR datasets is the camera setup used to capture the images or video sequences. We can distinguish between fixed-camera and moving-camera setups; obviously, this choice again depends on the application domain into which the HAR system will be integrated. For automotive applications or robotics, one should opt for a moving camera, as the camera movement might influence the visual properties of the human silhouettes. An example of a moving-camera dataset is Parse27k dataset [161]. For surveillance applications, a static camera setup will suffice. In another way, we could distinguish between indoor or outdoor camera setups; for example, RAP dataset [153] uses an indoor camera, while Parse27k dataset [161] comprises outdoor video sequences. Indoor captured datasets, such as [153], although captured in real-world scenarios, do not pose that many challenges as outdoor captured datasets, where the weather and lighting conditions are more volatile. Finally, the last aspect regarding the camera setup is related to the presence of a photographer. If the images are captured by a (professional) photographer some bias is introduced, as a human decides how and when to capture the images, such that it

will enhance the appearance of the subject. Some databases, such as CAD [123] or HAT [66,78] use images downloaded from public websites. However, in these images, the persons are aware of being photographed and perhaps even prepared for this (posing for the image, dressed up nicely for a photo session, etc.). Therefore, even if some datasets contain *in-the-wild* images gathered for a different system, they might still contain important differences from *real-world* images in which the subject is unaware of being photographed, the image is captured automatically, without any human intervention, are the subjects are dressed normally and performing natural dynamic movements.

6.  Pose and occlusion labeling. Another nice to have feature for a HAR dataset is the annotation of pose and occlusions. Some databases already provide this information [66,78,108,153]. Amongst other things, these extra labels prove useful in the evaluation of HAR systems, as they allow researchers to diagnose the errors of HAR and examine the influence of various factors.

7.  Data partitioning strategies. When dealing with HAR, the datasets partitioning scheme (into the train, validation, and test splits) should be carefully engineered. A common pitfall is to split the frames into the train and validation splits randomly, regardless of the person's identity. This can lead to an unfair assignment of a subject into one of these splits, and inducing bias in the evaluation process. This is even more important, as the current state-of-the-art methods generally rely on deep neural network architectures, which have a black-box behavior in nature, and it is not so straightforward to determine which image features lead to the final classification result.

    Solutions to this problem include extracting each individual (along with its track-lets) from the video sequence or providing the annotations at the identity level. Then, each person could be randomly assigned to one of the dataset splits.

8.  Synthetic data. Recently, significant advances have been made in the field of computer graphics and synthetic data generation. For example, in the field of drone surveillance, generated data [173] has proven its efficiency in training accurate machine vision systems. In this section, we have presented some computer-generated datasets which contain human attribute annotations. We consider that synthetically generated data is worth taking into consideration, as theoretically, it can be considered an inexhaustible source of data, which could be able to generate subjects with various attributes, under different poses, in diverse scenarios. However, state-of-the-art generative models rely on deep learning, which is known to be "hungry" for data, so data is needed to build a realistic generative model. Therefore, this solution might prove to be just a vicious circle.

9.  Privacy issues. In the past, as traditional video surveillance systems were simple and involved only human monitoring, privacy was not a major concern; however, these days, the pervasiveness of systems equipped with cutting-edge technologies in public places (e.g., shopping malls, private and public buildings, bus and train stations) have aroused new privacy and security concerns. For instance, Privacy Commissioner of Canada (OPC) is an organization that helps people report their privacy concerns and enforces the enterprises to manage people's personal data in their business activities based on restricting standards (https://www.priv.gc.ca/en/report-a-concern/).

    When gathering a dataset with real-world images, we deal with privacy and human rights violations. Ideally, HAR datasets should contained images captured by real-world surveillance cameras, with the subjects are unaware of being filmed, such that their behavior is as natural as possible. From an ethical perspective, humans should consent before their images are annotated and publicly distributed. However, this is not feasible for all scenarios. For example, *BRAINWASH* [174] dataset was gathered inside a private cafe for the purpose of head detection, and comprised 11,917 images. Although this benchmark is not very popular, it is seen in the lists of the popular datasets for commercial and military applications, as it has captured the regular customers without their awareness. *DUKE MTMC* [152] dataset targets the task of multi-person

re-identification from full-body images taken by several cameras. This dataset was collected in a university campus in an outdoor environment and contains over 2 million frames of 2000 students captured by 8 cameras at 1080p. *MS-CELEB-1M* [175] is another large dataset of 10 million faces collected from the Internet.

However, despite the success of these datasets (if we evaluate success by the number of citations and database downloads), the authors decided to shout-down the datasets due to human rights and privacy violation issues.

According to Pew Research Center Privacy Panel Survey conducted from 27 January to 16 February 2015, among 461 adults, more than 90 percent agreed that two factors are critical for surveillance systems: (1) *who* can access to their information? (2) *what* information is collected about them? Moreover, it is notable that they consent to share confidential information with someone they trust (93%); however, it is important not to be monitored without permission (88%).

As people's faces contain sensitive information that could be captured in the wild, authorities have published some standards (https://gdpr-info.eu/) to enforce enterprises respect the privacy of their costumers.

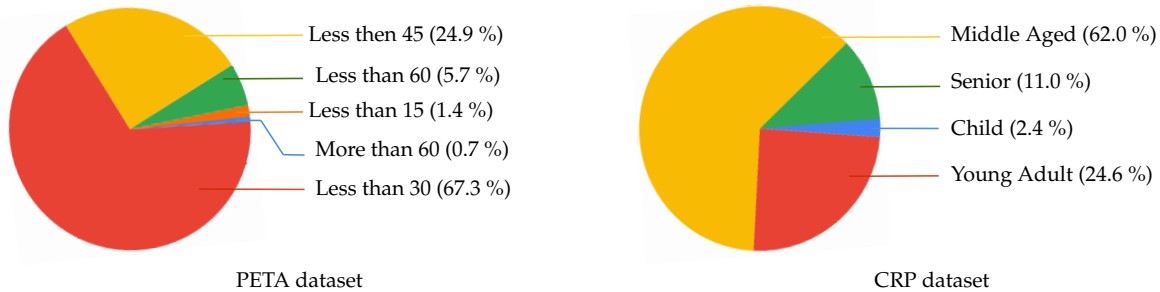

PETA dataset                                    CRP dataset

**Figure 4.** Frequency distribution of the labels describing the 'Age' class in the PETA [152] (on the **left**) and CRP [162] (on the **right**) databases.

### 6.2. Critical Discussion and Performance Comparison

As mentioned, the main objective of the localization method is to extract distinct fine-grained features, by careful analyses of different pieces of the input data and aggregating them. Although the extracted localized features create a detailed feature representation of the image, dividing the image to several pieces has several drawbacks:

- the expressiveness of the data is lost (e.g., when processing a jacket only by several parts, some global features that encode the jacket's shape and structure are ignored).
- as the person detector cannot always provide aligned and accurate bounding boxes, rigid partitioning methods are prone to error in body-part captioning, mainly when the input data includes a wide background. Therefore, methods based on stride/grid patching of the image are not robust to misalignment errors in the person bounding boxes, leading to degradation in prediction performance.
- different from gender and age, most human attributes (such as glasses, hat, scarf, shoes, etc.) belong to small regions of the image; therefore, analyzing other parts of the image may add irrelevant features to the final feature representation of the image.
- some attributes are view-dependent and highly changeable due to human body-pose, and ignoring them reduces the model performance; for example, glasses recognition in the side-view images is more laborious than front-view, while it may be impossible in back-view images. Therefore, in some localization methods (e.g., pose-let based techniques), regardless of this fact, features of different parts may be aggregated to perform a prediction on an unavailable attribute.

- some localization methods rely on the body-parsing techniques [176] or body-part detection methods [177] to extract local features. Not only requires training such part detectors rich annotations of data but also errors in body-parsing and body-part detection methods directly affect the performance of the HAR model.

There are several possibilities to address some of these issues, which mostly attempt to guide the learning process using additional information. For instance, as discussed in Section 3, some works use novel model structures [72] to capture the relationships and correlations between the parts of the image, while others try to use prior body-pose coordinates [63] (or develop a view-detector in the main structure [61]) to learn the view-specific attributes. Some methods develop attention modules to find the relevant body parts, while some approaches extract various pose-lets [163] of the image by slicing-window detectors. Using the semantic attributes as a constraint for extracting the relevant regions is another solution to look for localized attributes [100]. Moreover, developing accurate body-part detectors, body-parsing algorithms and introducing datasets with part annotations are some strategies that can help the localization methods.

Limited data is the other main challenge in HAR. The primary solutions for solving the problem of limited data are synthesizing artificial samples or augmenting the original data. One of the popular approaches for increasing the size of the dataset is to use generative models (i.e., Generative Adversarial Network (GAN) [178], Variational Auto-Encoders (VAE) [179], or a combination of both [180]). These models are powerful tools for producing new samples, but are not widely used for extending human full-body datasets for three reasons:

- in opposition to the breakthrough in face generative models [181], full-body generative models are still in early stages and their performance is still unsatisfactory,
- the generated data is unlabelled, while HAR is yet far from the stage to be implemented based on unlabeled data. It worth mentioning that, automatic annotations is an active research area in object detection [182].
- not only takes learning high-quality generative models for human full-body too much time, but it also requires a large amount of high-resolution learning data, which is yet not available.

Therefore, researchers [71,82,103,129,183–185] mostly either perform transfer learning to capture the useful knowledge of large datasets or resort to the simple yet useful label-persevering augmentation techniques from basic data augmentation (flipping, shifting, scaling, cropping, resizing, rotating, shearing, zooming, etc.) to more sophisticated methods such as random erasing [186] and foreground augmentation [187].

Due to the lack of sufficient data in some data classes (attributes), augmentation methods should be implemented carefully. Suppose that we have very few data from some classes (e.g., 'age 0–11', 'short winter jacket') and much more data from other classes (e.g., 'age 25–35', 't-shirt'). A blind data augmentation process would exacerbate the data class imbalance and increase the over-fitting problem in minority classes. Furthermore, some basic augmentations are not label persevering. For example, for a dataset annotated for body weight, scratching the images of a thin person may be interpreted as a medium or fat person, while it may be acceptable for color-based labels. Therefore, visualizing a set of augmented data and careful studying of the annotation data are highly suggested before performing augmentation.

Using *proper* pre-trained models (transfer learning) not only reduces the training time but also increases the system's performance. To have an effective transfer learning from task *A* to task *B* we should consider the following conditions [188–190],:

1. There should be some relationships between the data of task *A* and task *B*. For example, applying pre-trained weights of the ImageNet dataset [50] on HAR task is beneficial as both domains are dealing with RGB images of objects, including human data, while transferring the knowledge of

medical imagery (e.g., CT/MRI) are not useful and may only impose some heavy parameters to the model.

2. The data in task *A* is much more than the data in task *B* as transferring the knowledge of other small datasets cannot guarantee performance improvements.

Generally, there are two useful strategies for applying transfer learning to HAR problems, in which we suggest to discard the classification layers (i.e., fully connected layers that are on top of the model), and use the pre-trained model as a feature extractor (backbone). Then,

- we can freeze the backbone model and adding several classification layers on top of the model for fine-tuning.
- we can add the proper classification layers on top of the model and train all the model layers in several steps: (1) freeze the backbone model and fine-tune the last layers, (2) considering a lower learning rate, we unfreeze high-level feature extractor layers and fine-tune the model, (3) we unfreeze mid-level and low-level layers in other steps and train them with a lower learning rate, as these features are normally common between most tasks with the same data types.

Considering attribute correlations can boost the performance of HAR models. Some works (e.g., multi-task and RNN based models) attempt to extract the semantic relationship between the attributes from the visual data. However, lack of enough data and also the type of annotations in HAR datasets (the region of attributes are not annotated) lead to the poor performance of these models in capturing the correlation of attributes. Even in GCN and CRF based models that are known to be effective in capturing the relationship between defined nodes, yet these are no explicit mathematical expressions about several aspects: what is the optimal way to convert the visual data to some nodes, and what is the optimum number of nodes? When fusing the visual features with correlation information, how much should we give importance to the correlation information? How would be the performance of a model if it learns the correlation between attributes from external text data (particularly from the aggregation of several HAR datasets)?

Although occlusion is a primary challenge, yet few studies address it in HAR data. As surveyed in Section 3.4, several works have proposed to use video data, which is a rational idea only if more data are available. However, in still images, we know that even if most parts of the body (and even face of a person) is occluded, as human, we still are able to easily decid about many attributes of the person (see Figure 5). Another idea that could be considered in HAR data is labeling a/an (occluded) person with certain labels that are not correct. For example, suppose that the input data is a person with legging, even if the model is not certain about the correct lower body clothes, yet it could yield some labels indicating that the lower body cloth is not certainly a dress/skirt. Later, this information could be beneficial when considering the correlation between attributes. Moreover, introducing a HAR dataset composed of different degrees of occlusion could trigger more domain-specific studies. In the context of person re-id, Reference [191] provided an occluded dataset based on DukeMTMC dataset, which is not publicly available anymore (https://megapixels.cc/duke_mtmc/).

Last but not least, studies based on class imbalance challenge attempt to promote the importance of the minority classes and (or) decrease the importance of majority classes, by proposing hard and (or) soft solutions. As mentioned earlier, providing more data blindly (collecting or augmenting the existing data) cannot guarantee better performance and may increase the gap between the number of samples in data classes. Therefore, we should provide a trade-off between the downsampling and upsampling strategies, while using the proper loss functions to learn more from minority samples. As discussed in Section 3.5, these ideas have been developed to some extent; however, other challenges in HAR have been neglected in the final proposal.

**Negative attributes**

Lower body clothes:
    **- skirt, pants, bikini**

Shoes:
    **- slippers, bare foot**

**Positive attributes**

**+ young,**
**+ female,**
**+ sports bag,**
**+ winter jacket,**
**+ pony tail,**
**+ long hairstyle**

**Figure 5.** As human, not only we describe the available attributes in occluded images but also we can predict the covered attributes in a negative strategy based on the attribute relations.

Table 2 shows the performance of the HAR approaches over the last decade and indicates a consistent improvement of methods over time. In 2016, the *F*1 performance evaluation of [74] on the RAP and PETA datasets was 66.12 and 84.90, respectively, while these numbers were improved to 79.98 and 86.87 in the year 2019 [92] and to 82.10 and 88.30 in year 2020. Furthermore, according to Table 2, it is clear that challenges of attributes localization and attributes correlation have attracted the most attention over the recent years, which indicates that extracting distinctive fine-grained features from relevant locations of the given input images is the most important aspect of HAR models.

Despite the early works that analyzed the human full-body data in different locations and situations, recent works have focused on attribute recognition from surveillance data, which arouses some privacy issues.

Appearing comprehensive evaluation metrics is another noticeable change over the last decade. Due to the intrinsic, large class imbalance in the HAR datasets, $mA$ cannot provide a comprehensive performance evaluation over different methods. Suppose that in a binary classification situation, if 99% of the samples belong to persons with glasses and 1% of samples belong to persons without glasses, the model can recognize all the test samples as persons with glasses and still has 99% of accuracy in recognition. Therefore, for a fair performance comparison with the state of the arts, it is necessary to consider metrics such as *Prec*, *Rec*, *Acc*, and *F*1 – which are discussed in Section 5.

Table 2 also shows that the RAP, PETA, and PA-100K datasets have attracted the most attention in the context of attribute recognition –which excludes person re-id. In Figure 6 we illustrate the state-of-the-art results obtained on these datasets for $mAP$ metric. As seen, the PETA dataset sounds easier than other datasets, despite the smaller size and lower quality data compared with the RAP dataset.

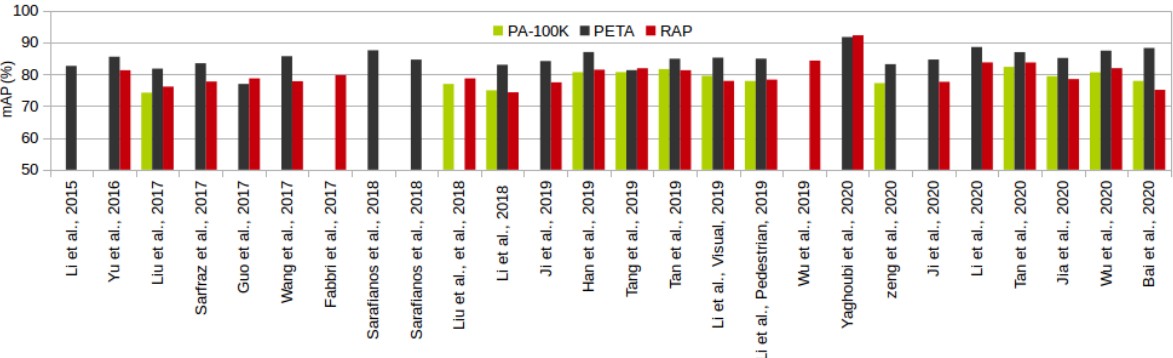

**Figure 6.** State-of-the-art mAP results on three well-known PAR datasets.

**Table 2.** Performance comparison of HAR approaches over the last decade for different benchmarks.

| Ref., Year, Cat. | Taxonomy | Dataset | mA | Acc. | prec. | rec. | F1 |
|---|---|---|---|---|---|---|---|
| [66], 2011, FAR | Pose-Let | HAT [66] | 53.80 | - | - | - | - |
| [68], 2011, FAR | Pose-Let | [68] | 82.90 | - | - | - | - |
| [123], 2012, FAR and CAA | Attribute relation | [123] | - | 84.90 | - | - | - |
| | | D.Fashion [91] | 35.37 (top-5) | - | - | - | - |
| [79], 2013, FAR | Body-Part | HAT [66] | 69.88 | - | - | - | - |
| [69], 2013, FAR | Pose-Let | HAT [66] | 59.30 | - | - | - | - |
| [70], 2013, FAR | Pose-Let | HAT [66] | 59.70 | - | - | - | - |
| [77], 2015, FAR | Body-Part | DP [68] | 83.60 | - | - | - | - |
| [128], 2015, PAR | Loss function | PETA [152] | 82.6 | - | - | - | - |
| [77], 2015, FAR | Body-Part | DP [68] | 83.60 | - | - | - | - |
| [150], 2015, CAA | Attribute location and relation | Dress [150] | - | 84.30 | 65.20 | 70.80 | 67.80 |
| [75], 2016, FAR | Pose-Let | WIDER [75] | 92.20 | - | - | - | - |
| [74], 2016, PAR | Pose-Let | RAP [108] | 81.25 | 50.30 | 57.17 | 78.39 | 66.12 |
| | | PETA [152] | 85.50 | 76.98 | 84.07 | 85.78 | 84.90 |
| [91], 2016, CAA | Limited data | D.Fashion [91] | 54.61 (top-5) | - | - | - | - |
| [86], 2017, FAR | Attention | WIDER [75] | 82.90 | - | - | - | - |
| | | Berkeley [68] | 92.20 | - | - | - | - |
| [88], 2017, PAR | Attention | RAP [108] | 76.12 | 65.39 | 77.33 | 78.79 | 78.05 |
| | | PETA [152] | 81.77 | 76.13 | 84.92 | 83.24 | 84.07 |
| | | PA-100K [88] | 74.21 | 72.19 | 82.97 | 82.09 | 82.53 |
| [124], 2018, FAR | Grammar | DP [68] | 89.40 | - | - | - | - |
| [61], 2018, PAR and FAR | Pose Estimation | RAP [108] | 77.70 | 67.35 | 79.51 | 79.67 | 79.59 |
| | | PETA [152] | 83.45 | 77.73 | 86.18 | 84.81 | 85.49 |
| | | WIDER [75] | 82.40 | - | - | - | - |
| [86], 2017, PAR | Attention | RAP [108] | 78.68 | 68.00 | 80.36 | 79.82 | 80.09 |
| | | PA-100K [88] | 76.96 | 75.55 | 86.99 | 83.17 | 85.04 |
| [109], 2017, PAR | RNN | RAP [108] | 77.81 | - | 78.11 | 78.98 | 78.58 |
| | | PETA [152] | 85.67 | - | 86.03 | 85.34 | 85.42 |
| [104], 2017, PAR | Loss Function - Augmentation | PETA [152] | - | 75.43 | - | 70.83 | - |
| [132], 2017, PAR | Occlusion | RAP [108] | 79.73 | 83.97 | 76.96 | 78.72 | 77.83 |
| [105], 2017, CAA | Transfer Learning | [105] | 64.35 | - | 64.97 | 75.66 | - |
| [111], 2017, PAR | Multitask | Market [24] | - | 88.49 | - | - | - |
| | | Duke [152] | - | 87.53 | - | - | - |
| [192], 2017, CAA | Multiplication | D.Fashion [91] | 30.40 (top-5) | - | - | - | - |
| [89], 2018, CAA | Attention | D.Fashion [91] | 60.95 (top-5) | - | - | - | - |
| [112], 2018, PAR | Soft-Multitask | SoBiR [114] | 74.20 | - | - | - | - |
| | | VIPeR [193] | 84.00 | - | - | - | - |
| | | PETA [152] | 87.54 | - | - | - | - |

**Table 2.** *Cont.*

| Ref., Year, Cat. | Taxonomy | Dataset | mA | Acc. | prec. | rec. | F1 |
|---|---|---|---|---|---|---|---|
| [149], 2018, PAR and FAR | Soft solution | WIDER [75] | 86.40 | - | - | - | - |
| | | PETA [152] | 84.59 | 78.56 | 86.79 | 86.12 | 86.46 |
| [97], 2018, PAR | Attribute location | RAP [108] | 78.68 | 68.00 | 80.36 | 79.82 | 80.09 |
| | | PA-100K [88] | 76.96 | 75.55 | 86.99 | 83.17 | 85.04 |
| [63], 2018, PAR | Pose Estimation | PETA [152] | 82.97 | 78.08 | 86.86 | 84.68 | 85.76 |
| | | RAP [108] | 74.31 | 64.57 | 78.86 | 75.90 | 77.35 |
| | | PA-100K [88] | 74.95 | 73.08 | 84.36 | 82.24 | 83.29 |
| [117], 2018, PAR | RNN | RAP [108] | - | 77.81 | 78.11 | 78.98 | 78.58 |
| | | PETA [152] | - | 85.67 | 86.03 | 85.34 | 85.42 |
| [126], 2019, PAR | Soft solution | RAP [108] | 77.44 | 65.75 | 79.01 | 77.45 | 78.03 |
| | | PETA [152] | 84.13 | 78.62 | 85.73 | 86.07 | 85.88 |
| [125], 2019, PAR | Multiplication | PETA [152] | 86.97 | 79.95 | 87.58 | 87.73 | 87.65 |
| | | RAP [108] | 81.42 | 68.37 | 81.04 | 80.27 | 80.65 |
| | | PA-100K [88] | 80.65 | 78.30 | 89.49 | 84.36 | 86.85 |
| [118], 2019, PAR | RNN | RAP [108] | - | 77.81 | 78.11 | 78.98 | 78.58 |
| | | PETA [152] | - | 86.67 | 86.03 | 85.34 | 85.42 |
| [133], 2019, PAR | Occlusion | Duke [152] | - | 89.31 | - | - | 73.24 |
| | | MARS [194] | - | 87.01 | - | - | 72.04 |
| [100], 2019, PAR | Attribute Location | RAP [108] | 81.87 | 68.17 | 74.71 | 86.48 | 80.16 |
| | | PETA [152] | 86.30 | 79.52 | 85.65 | 88.09 | 86.85 |
| | | PA-100K [88] | 80.68 | 77.08 | 84.21 | 88.84 | 86.46 |
| [92], 2019, PAR | Attention | PA-100K [88] | 81.61 | 78.89 | 86.83 | 87.73 | 87.27 |
| | | RAP [108] | 81.25 | 67.91 | 78.56 | 81.45 | 79.98 |
| | | PETA [152] | 84.88 | 79.46 | 87.42 | 86.33 | 86.87 |
| | | Market [24] | 87.88 | - | - | - | - |
| | | Duke [152] | 87.88 | - | - | - | - |
| [110], 2019, PAR | GCN | RAP [108] | 77.91 | 70.04 | 82.05 | 80.64 | 81.34 |
| | | PETA [152] | 85.21 | 81.82 | 88.43 | 88.42 | 88.42 |
| | | PA-100K [88] | 79.52 | 80.58 | 89.40 | 87.15 | 88.26 |
| [107], 2019, PAR | GCN | RAP [108] | 78.30 | 69.79 | 82.13 | 80.35 | 81.23 |
| | | PETA [152] | 84.90 | 80.95 | 88.37 | 87.47 | 87.91 |
| | | PA-100K [88] | 77.87 | 78.49 | 88.42 | 86.08 | 87.24 |
| [94], 2019, PAR and FAR | Attention | RAP [108] | 84.28 | 59.84 | 66.50 | 84.13 | 74.28 |
| | | WIDER [75] | 88.00 | - | - | - | - |
| [96], 2020, PAR | Attention | RAP [108] | 92.23 | - | - | - | - |
| | | PETA [152] | 91.70 | - | - | - | - |
| [54], 2020, PAR | Multi-task | PA-100K [88] | 77.20 | 78.09 | 88.46 | 84.86 | 86.62 |
| | | PETA [152] | 83.17 | 78.78 | 87.49 | 85.35 | 86.41 |
| [195], 2020, PAR | RNN | RAP [108] | 77.62 | 67.17 | 79.72 | 78.44 | 79.07 |
| | | PETA [152] | 84.62 | 78.80 | 85.67 | 86.42 | 86.04 |
| [58], 2020, PAR | RNN and attention | RAP [108] | 83.72 | - | 81.85 | 79.96 | 80.89 |
| | | PETA [152] | 88.56 | - | 88.32 | 89.62 | 88.97 |

**Table 2.** *Cont.*

| Ref., Year, Cat. | Taxonomy | Dataset | mA | Acc. | prec. | rec. | F1 |
|---|---|---|---|---|---|---|---|
| [120], 2020, PAR | GCN | RAP [108] | 83.69 | 69.15 | 79.31 | 82.40 | 80.82 |
| | | PETA [152] | 86.96 | 80.38 | 87.81 | 87.09 | 87.45 |
| | | PA-100K [88] | 82.31 | 79.47 | 87.45 | 87.77 | 87.61 |
| [196], 2020, PAR | Baseline | RAP [108] | 78.48 | 67.17 | 82.84 | 76.25 | 78.94 |
| | | PETA [152] | 85.11 | 79.14 | 86.99 | 86.33 | 86.09 |
| | | PA-100K [88] | 79.38 | 78.56 | 89.41 | 84.78 | 86.25 |
| [59], 2020, PAR | RNN and attention | PA-100K [88] | 80.60 | - | 88.70 | 84.90 | 86.80 |
| | | RAP [108] | 81.90 | - | 82.40 | 81.90 | 82.10 |
| | | PETA [152] | 87.40 | - | 89.20 | 87.50 | 88.30 |
| | | Market [24] | 88.50 | - | - | - | - |
| | | Duke [152] | 88.80 | - | - | - | - |
| [197], 2020, PAR | Hard solution | PA-100K [88] | 77.89 | 79.71 | 90.26 | 85.37 | 87.75 |
| | | RAP [108] | 75.09 | 66.90 | 84.27 | 79.16 | 76.46 |
| | | PETA [152] | 88.24 | 79.14 | 88.79 | 84.70 | 86.70 |
| [198], 2020, CAA | — | Fashionista [199] | - | 88.91 | 47.72 | 44.92 | 39.42 |
| [200], 2020, PAR | Math-oriented | Market [24] | 92.90 | 78.01 | 87.41 | 85.65 | 86.52 |
| | | Duke [152] | 91.77 | 76.68 | 86.37 | 84.40 | 85.37 |

We observe that the performance of the state of the arts is yet far from the reliable range to be used in forensic affairs and enterprises, and it requires more attention in both introducing novel datasets and proposing robust methods.

Among PAR, FAR, and CAA fields of study, most of the papers have focused on the PAR task. The reason is not apparent, but at least we know that (1) PAR data are often collected from CCTV and surveillance cameras, and analyzing such data is critical for forensic and security objectives, (2) person re-id is a hot topic that mainly works with the same data type and could be highly influenced by powerful PAR methods.

## 7. Conclusions

This survey reviewed the most relevant works published in the context of human attributes recognition problem (HAR) over the last decade. Contrary to the previous published reviews, which provided a methodological categorization of the literature, in this survey we privileged a challenge-based taxonomy, that is, methods were organized based on the challenges of the HAR problem that they were devised to address. According to this type of organization, readers can easily understand the most suitable strategies for addressing each of the typical challenges of HAR and simultaneously learn which strategies perform better. In addition, we comprehensively reviewed the available HAR datasets, outlining the relative advantages and drawbacks of each one with respect to others, as well as the data collection strategy used. Also, the intrinsically imbalanced nature of the HAR datasets is discussed, as well the most relevant challenges that typically arise when gathering data for this problem.

**Author Contributions:** E.Y.'s contributions are in conceptualization, methodology, formal analysis, data curation, and writing the original draft. F.K. collaborated in conceptualization, methodology, and formal analysis. D.B. collaborated in writing, conceptualization, data curation, and formal analysis of Sections 4 and 1 and reviewed the manuscript. S.A.K. collaborated in formal analysis and in the data curation of Section 6.2. J.N. collaborated in reviewing, editing the manuscript, and funding acquisition. H.P. is the supervisor of the project and performed the project administration, conceptualization, funding acquisition, and manuscript revision. All authors have read and agreed to the published version of the manuscript.

**Funding:** This research is funded by the "Fundo Europeu de Desenvolvimento Regional (FEDER), Fundo de Coesao (FC) and Fundo Social Europeu (FSE)" under the "PT2020 - Portugal 2020" Program, "IT: Instituto de Telecomunicações" and "TOMI: City's Best Friend" with reference POCI-01-0247-FEDER-033395. Also, the work is funded by Fundação para a Ciência e a Tecnologia / Ministério da Educação e Ciência (FCT/MEC) through national funds and, when applicable, co-funded by the FEDER PT2020 partnership agreement under the project UID/EEA/50008/2019).

**Conflicts of Interest:** The authors declare no conflict of interest. The funders had no role in the design of the study; in the collection, analyses, or interpretation of data; in the writing of the manuscript, or in the decision to publish the results.

## Abbreviations

The following abbreviations are used in this manuscript:

| | |
|---|---|
| Acc | Accuracy |
| CAA | Clothing Attribute Analysis |
| CAMs | Class Activation Maps |
| CCTV | Closed-Circuit TeleVision |
| CNN | Convolutional Neural Network |
| CSD | Color Structure Descriptor |
| CRF | Conditional Random Field |
| DPM | Deformable Part Model |
| FAA | Facial Attribute Analysis |
| FAR | Full-body Attribute Recognition |
| FP | False Positives |
| GAN | Generative Adversarial Network |
| GCN | Graph Convolutional Network |
| HAR | Human Attribute Recognition |
| HOG | Histogram of Oriented Gradients |
| re-id | re-identification |
| LMLE | Large Margin Local Embedding |
| LSTM | Long Short Term Memory |
| mA | mean Accuracy |
| MAP | Maximum A Posterioris |
| MAResNet | Multi-Attribute Residual Network |
| MCSH | Major Colour Spectrum Histogram |
| MSCR | Maximally Stable Colour Regions |
| Prec | Precision |
| Rec | Recall |
| ResNet | Residual Networks |
| RHSP | Recurrent Highly-Structured Patches |
| RoI | Regions of Interests |
| RNN | Recurrent Neural Networks |
| SIFT | Scale Invariant Feature Transform |
| SE-Net | Squeeze-and-Excitation Networks |
| SMOTE | Synthetic Minority Over-sampling TEchnique |
| SPR | Spatial Pyramid Representation |
| SSD | Single Shot Detector |
| STN | Spatial Transformer Network |
| SVM | Support Vector Machine |
| TN | True Negative |
| TP | True Positive |
| UAV | Unmanned Aerial Vehicle |
| VAE | Variational Auto-Encoders |
| YOLO | You Only Look Once |

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
