# Peer review of "Human Attribute Recognition— A Comprehensive Survey"

_applsci, doi:10.3390/app10165608_

Round 1

Reviewer 1 Report

This paper presents a literature review for recognition of human attributes such as gender, age, clothing, etc. in pedestrian settings.

Although it proposes a taxonomy to analyze the main challenges and summarizes relevant literature, the paper does not critically examine the contributions of past research, explain the results of past research nor clarify alternative views.
In other words, the review lacks the critical interpretation of the authors of the literature.

In addition, the problems that are analyzed in the review are general problems in artificial vision or in other machine learning domains.
I would like to see in the paper an analysis of what makes the problem hard in this particular domain, what methods can be domain-agnostic and what methods should be or are domain specific.
Is there some knowledge in the domain that helps in the recognition? Are there some problems unique to this domain?
There is a small discussion section that addresses these concerns, it would be beneficial to expand and make it the main section of the paper.

Finally, there have been on-going conversations about the ethics of similar systems that are being used as surveillance systems and to recognize people in protests. I think it is important to have a discussion in the paper about the ethical implications of the approaches and their biases.

As a minor suggestion, HAR is commonly used for Human Activity Recognition, I would suggest looking for potential misleading.

Author Response

The review letter has been attached as a PDF file.

Reviewer 2 Report

Dear Authors,

congratulations for your effort and for the proficient survey analysis. The effort of providing a strategic approach to HAR is much needed and specific for each field of application. The latter was represented widely in the manuscript and enriched by a strong dataset review and solid data collection.

Author Response

(The authors gave the same response as above.)

Reviewer 3 Report

This paper presents a valid overview of the current available datasets and implemented deep learning solutions in the Human Attribute Recognition. As stated by the authors, it mainly focuses on three of out 4 subsections of this field. Namely, facial attribute analysis is out of the scope of this work. 

Pros:

  • It provides a deep description of the available datasets in literature
  • It provides the identification of the main challenges on the recognition task as well as some solutions present in the current literature. 

Cons: 

  • There is no clear comparison among the techniques described in each subsection of Section 2. It would enrich the quality of the paper and help the reader in going through it. It could be down also by inserting a table with summarise the literature reviewed.
  • The paper is well-written, but relevant references on some statements are missing(i.e. on the definition of "human attribute recognition", on the factors listed at the end of section 1, on the definition of Attention heat maps, etc). 
  • On the structure of the work, the description of the dataset and of the metrics should be put before the description of the learning methods, since both metrics and dataset are recalled in the section. 
  • On the dataset description, it is no clear where the number of Figure 3 and Figure 4 comes from. 
  • Some issues presented in Section 1 and 2 are shared among several recognition tasks, not only human attribute. A more clear explanation in that sense should be provided. 

Author Response

(The authors gave the same response as above.)

Reviewer 4 Report

This paper presents a survey on Human Attribute Recognition. The topic is relevant, the paper is timely and well written. However, there are some aspects that need to be thoroughly improved, specially given the fact that this is a survey paper. 

First of all, there is very little introduction of what HAR is. The authors briefly describe it in a paragraph in the introduction and the section 1.1. But, this description is very limited and not very tutorial like. This makes the paper to be directed to experts on the topic, rather than to a less expert reader who could enhance the knowledge on the topic thanks to this paper. To improve this, I would suggest the authors to add a Section 2 in which the HAR principles are very well introduced and the importance of machine learning in the topic is discussed. In its current form, nearly all the presented approaches are machine learning-based, but nowhere is explained why this is the case. Is it not possible to HAR without Machine Learning?

Second, the driving motivation to provide this survey is rather unclear. Some comparing notes with other surveys are provided but the motivation is rather missing. One key reason for this is the fact that the introduction does not really provide an introduction to the topic, starting to why HAR is needed (big picture) to the contributions of the paper. I would suggest the authors to look at introductions of other papers to get inspired by their structure. The same goes for the abstract, which currently provides only a summary of the paper, but rather avoids to present the motivation of the paper.

Third, the paper seems to provide a summary of different approaches, but not much of a discussion of lessons learned. It is true that sections 2.7 and 3.4 present some conclusions. However, I believe a more general discussion of the paper would highly increase the quality of the paper. I would suggest the authors to move sections 2.7 and 3.4 to a new section (section 4) called discussion. In it, the table 1 could be enhanced with the evaluation method as well as the sub-category of the area of human attribute analysis (which seems to be quite ignored throughout the paper). And further discussion could be provided.

Finally, some minor things:

  1. The subsections in section 2 are in a counter-intuitive order taking into account Figure 2. I would suggest the authors to either use the order of the figure or update the figure to follow the order of the sections.
  2. The challenges in section 2 should be clearly explained before jumping into the state of the art approaches.
  3. The draft has some typos and grammatical mistakes. Some examples:
    1. increasing amounts -> increasing amount 
    2. facial attribute analysis aims at is at -> aims at
    3. kinds of annotations -> types of annotations

Author Response

(The authors gave the same response as above.)

Round 2

Reviewer 3 Report

The paper shows evident improvements. 

I think it could be accepted in the present form. 

Reviewer 4 Report

The authors have addressed all my comments. I have no further comments.